# *Mycoplasma bovis* Infection Induces Apoptosis Through Gadd45/XIAP in Bovine Macrophages

**DOI:** 10.3390/microorganisms13092031

**Published:** 2025-08-30

**Authors:** Ruirui Li, Xiaojiao Yu, Tian Tang, Jinliang Sheng, Hui Zhang, Chuangfu Chen, Yong Wang, Zhongchen Ma

**Affiliations:** College of Animal Science and Technology, Shihezi University, Shihezi 832000, China; 20242013058@stu.shzu.edu (R.L.); yuxiaojiao@stu.shzu.edu.cn (X.Y.); liangjiayin@stu.shzu.edu.cn (T.T.); wangwenjia@stu.shzu.edu.cn (J.S.); zhangqianyi@stu.shzu.edu.cn (H.Z.); ccf@shzu.edu.cn (C.C.)

**Keywords:** *Mycoplasma bovis*, transcriptome, apoptosis, Gadd45, XIAP

## Abstract

*Mycoplasma bovis* (*M. bovis*) adheres to host cells and persists intracellularly, causing chronic inflammation and significant economic losses in the cattle industry. The role of host cell apoptosis in this host–pathogen interaction remains unclear. This study isolated and identified the *M. bovis* Xinjiang strain XJ01 from diseased cattle in China. XJ01 exhibited typical “fried egg” colony morphology, distinct biochemical characteristics, and a 1.02 Mb genome (29.33% GC content) encoding 939 genes, including 93 unique genes. Functional analysis under optimal infection conditions (MOI = 1000, 24 h) revealed that XJ01 induced significant apoptosis and reduced viability in bovine macrophages (BoMac). This was accompanied by mitochondrial homeostasis disruption, characterized by increased Bax expression and suppressed Bcl-2 levels. Transcriptome analysis identified 9926 differentially expressed genes. KEGG pathway enrichment indicated significant activation of apoptosis and P53 signaling pathways, with Gadd45 and XIAP identified as key regulators. Mechanistic validation demonstrated that Gadd45 overexpression or XIAP knockdown enhanced Bax expression, inhibited Bcl-2, increased apoptosis rates, and consequently significantly reduced intracellular bacterial load at 24 h post-infection. Conversely, suppressing Gadd45 or overexpressing XIAP promoted pathogen survival. Collectively, this study reveals that *M. bovis* XJ01 activates host stress signaling to upregulate Gadd45 and suppress XIAP, thereby triggering mitochondrial apoptosis as a mechanism to eliminate intracellular bacteria—illustrating a self-limiting antibacterial mechanism.

## 1. Introduction

*Mycoplasma bovis* (*M. bovis*), a cell wall-deficient pathogen, invades the bovine respiratory system and joint tissues. It is a primary causative agent of Bovine Respiratory Disease (BRD), leading to pneumonia, arthritis, and mastitis [1,2,3,4]. Since its initial isolation from a mastitic cow in 1961 and subsequent confirmation as a BRD pathogen in 1976 [5], *M. bovis* has achieved global distribution, causing persistent economic losses to the cattle industry [6,7]. Mycoplasma species possess the smallest genomes among prokaryotes (0.58–1.4 Mbp) [8], and their pathogenicity exhibits strain-specificity. Pathogenesis primarily involves immune evasion and intracellular colonization mediated by adhesin-mediated cellular invasion [9], release of toxic metabolites (e.g., hydrogen peroxide) [10,11], and secretion of exotoxins [12,13]. Notably, *M. bovis* displays significant geographic genetic diversity [14], and its virulence factors and host interaction modes frequently undergo dynamic evolution depending on the prevalent strain lineage. Xinjiang, an important animal husbandry region in China, has experienced frequent outbreaks of *M. bovis* infection in recent years. However, the molecular characteristics and pathogenic mechanisms of the prevalent strains in this region remain poorly characterized.

Apoptosis exhibits a dual regulatory characteristic in *M. bovis*-host interactions. On one hand, *M. bovis* can induce lymphocyte apoptosis [15] and trigger monocyte apoptosis by activating the nuclear factor kappa-light-chain-enhancer of activated B cells (NF-κB) pathway [16]. On the other hand, it impedes xenophagy by inhibiting host caspase activity, thereby promoting intracellular survival [17]. Of particular note, in macrophage infection models, the Mb1 strain significantly delays host apoptosis by markedly upregulating B-cell lymphoma 2 (Bcl-2) expression and inhibiting caspase-3, -6, and -9 activity [18,19]. This dual apoptosis-modulation strategy suggests that *M. bovis* may achieve immune evasion by targeting specific host genes to dynamically balance apoptotic pathways. Based on this paradoxical phenomenon, this study focuses on the core scientific question: How do prevalent Xinjiang strains of *M. bovis* mediate the dynamic interplay between pathogenicity and host defense through the regulation of host apoptotic pathways?

This study focuses on the isolation and identification of the *M. bovis* Xinjiang prevalent strain XJ01, and its interaction system with bovine macrophages (BoMac). By establishing an in vitro apoptosis model and integrating transcriptomic analysis, we reveal that the XJ01 strain specifically upregulates growth arrest and DNA damage-inducible 45 (Gadd45) expression while suppressing X-linked inhibitor of apoptosis protein (XIAP), thereby disrupting B-cell lymphoma 2 (Bcl-2)/Bcl-2-associated X protein (Bax) homeostasis and inducing the mitochondrial apoptosis pathway to restrict intracellular pathogen survival. This discovery not only elucidates a novel self-limiting antimicrobial mechanism but also provides a molecular theoretical basis for deciphering *M. bovis* immune evasion strategies.

## 2. Materials and Methods

### 2.1. Culture of Cells and M. bovis

The BoMac cells were kindly provided by Professor Guo Aizhen’s research team at Huazhong Agricultural University. BoMac cells were cultured in RPMI 1640 medium (Gibco, CA, USA) supplemented with 10% fetal bovine serum (FBS) (Gibco, CA, USA) and 1% penicillin-streptomycin (Solarbio, Beijing, China) at 37 °C in a humidified 5% CO_2_ incubator. The *M. bovis* isolate XJ01 was cultured at 37 °C in modified PPLO broth liquid medium containing 10% horse serum (HyClone, UT, USA). The PPLO broth medium comprised the following components: PPLO Broth Base (BD, California, USA), Sodium pyruvate (Sigma-Aldrich, Darmstadt, Germany), Yeast extract (BD, USA), 10% Horse serum (HyClone, UT, USA), Minimum Essential Medium (MEM) (Gibco, CA, USA), Penicillin (Solarbio, Beijing, China), Phenol red (Sinopharm Chemical Reagent Co., Ltd., Suzhou, China).

### 2.2. Isolation and Identification of M. bovis

During 2023–2024, 84 nasal mucus swabs were collected from symptomatic cattle exhibiting cough and nasal discharge across 8 farms in 8 cities of Xinjiang. Additionally, paired samples of lung and spleen tissues (*n* = 2 each) were obtained from deceased cattle with characteristic pathological lesions. All specimens were cryopreserved in Mycoplasma broth and transported to the laboratory promptly for processing.

Nasal swab samples and tissue homogenates were vortexed, centrifuged, and filtered through 0.45 μm membranes. The filtrates were supplemented with fresh culture medium and incubated at 37 °C under 5% CO_2_. Upon development of uniform turbidity (yellowish hue), samples were filtered again and subjected to three rounds of 1:10 serial dilution and passage culture. Bacterial suspensions were then plated on agar plates and cultured for 5 days. Typical “fried egg-shaped” colonies were selected for three cycles of solid–liquid alternating purification. Isolates were preserved in –80 °C glycerol stocks. Genomic DNA was extracted from purified strains, and UvrC and 16S rRNA gene fragments were amplified by PCR. After electrophoretic verification, 16S rRNA products were sequenced and aligned against the NCBI database for species confirmation. Concurrently, growth curves of isolates in PPLO broth were determined over 120 h using the color changing unit (CCU) method. Biochemical profiles (glucose, mannitol, gelatin hydrolysis, esculin hydrolysis, lactose, and triphenyl tetrazolium chloride reduction) were assessed, and antibiotic susceptibility was tested by disk diffusion assay.

### 2.3. Whole-Genome Sequencing of M. bovis Isolate XJ01

Genomic DNA of the isolate was submitted to Beijing Novogene Bioinformatics Technology Co., Ltd. (Beijing, China). for single-molecule real-time (SMRT) sequencing on the PacBio Sequel II platform. Library preparation was performed using the PacBio SMRTbell™ Template Kit, involving standard procedures: DNA fragmentation, magnetic bead size selection, end repair, damage repair, and hairpin adapter ligation. Raw sequencing data underwent quality control via the PacBio SMRT Link pipeline (removing low-quality reads and adapter sequences), followed by de novo assembly using Canu software (V.1.7.1 21). Genome annotation integrated GeneMarkS (gene prediction), RepeatMasker (repeat sequence identification), tRNAscan-SE (tRNA prediction), and the NCBI Prokaryotic Genome Annotation Pipeline (PGAP). Functional annotation was performed against KEGG, COG, and NR databases.

### 2.4. Comparative Genomics Analysis of M. bovis Isolate XJ01

This study selected eight reference *M. bovis* isolates for comparative genomic analysis alongside the experimental isolate. Reference strain genomes were downloaded from NCBI GenBank (details in Table 1). Pangenome analysis was conducted by clustering protein sequences across samples using cd-hit (v4.6.1), with subsequent visualization performed on the R platform (v3.2.4). Whole-genome alignment between the target and reference genomes was carried out using MUMmer (v3.23) to identify broad collinearity characteristics. Phylogenetic tree construction was implemented via Treebest (v1.9.2).

### 2.5. Cytotoxicity Assessment and Condition Optimization of M. bovis Infection in BoMac Cells

BoMac cells were seeded at 5×10^5^ cells per well in 6-well plates and incubated overnight at 37 °C with 5% CO_2_ until a confluent monolayer formed. *M. bovis* isolate XJ01 in logarithmic growth phase was inoculated into cell wells at an MOI of 1000. Infections proceeded for 6 h, 12 h, 24 h, 36 h, and 48 h alongside uninfected control groups. Cells were washed three times with ice-cold PBS followed by assessment of viability and apoptosis using CCK-8 and apoptosis assay kits.

To identify the optimal MOI, *M. bovis* isolate XJ01 in logarithmic growth phase was inoculated into cell wells at MOIs of 0.1, 1, 10, 100, and 1000. After 24 h of infection, alongside uninfected control groups, cells were washed three times with ice-cold PBS. Cell viability and apoptosis were then assessed using CCK-8 and apoptosis assay kits.

### 2.6. CCK8 Assay for Cell Viability

The CCK8 method was used to detect BoMac cell proliferation and viability. Blank wells (medium only without cells), control wells (cells with medium only), and experimental wells (cells with *M. bovis* and medium) were set up. In 96-well plates, 100 μL of cell suspension was added per well. After 12–24 h, 10 μL of CCK8 solution was added to each well, followed by incubation at 37 °C for 4 h. Absorbance was measured at 450 nm wavelength. Cell viability was calculated as: Cell Viability% = (OD_450nm_ experimental group–OD4_50nm_ blank group)/(OD_450nm_ control group–OD_450nm_ blank group), with three replicates per group.

### 2.7. Detection of Cell Apoptosis

Cell apoptosis rates were detected using the Annexin V-FITC/PI Apoptosis Detection Kit (Solarbio, Beijing, China). BoMac cells were infected with *M. bovis* at MOIs of 0.1, 1, 10, 100, and 1000 for 6 h, 12 h, 24 h, 36 h, and 48 h. Culture supernatants were collected into centrifuge tubes, and cells were digested with EDTA-free trypsin. The digested cells were transferred to corresponding centrifuge tubes containing the collected supernatants and centrifuged at 400× *g* for 5 min. After washing and pelleting with PBS, cells were resuspended in 100 μL of 1× Binding Buffer. Then, 5 μL Annexin V-FITC and 5 μL PI were added and mixed thoroughly, followed by incubation at room temperature in the dark for 8–10 min. Finally, the mixture was diluted with 400 μL of 1× Binding Buffer and vortexed for flow cytometry analysis (BD, California, USA).

### 2.8. Quantitative Real-Time PCR (qRT-PCR)

Total RNA was extracted from BoMac cells using Trizol (ComWin, Beijing, China) and reverse transcribed using the HiFiScript cDNA Synthesis Kit (ComWin, Beijing, China). Primers listed in Appendix A were utilized for reverse transcription, with the optimal interfering fragments selected based on primer efficiencies exceeding 70%. Quantitative detection employed the SYBR Green dye method, with reaction mixtures containing 5 μL SYBR Green qPCR Mix, 0.2 μL each of forward and reverse primers, 3.6 μL ddH_2_O, and 1 μL cDNA. Amplification was performed on a QuantStudio 5 Real-Time PCR System (Thermo Fisher, MA, USA) under the following program: initial denaturation at 95 °C for 5 s, followed by 45 cycles of 95 °C for 5 s (denaturation) and 60 °C for 30 s (annealing/extension). All samples were analyzed using this protocol. Data were normalized against GAPDH expression levels, with melt curve analysis confirming single amplicon products. Relative expression values were calculated using the 2^−ΔΔCT^ method and presented as fold changes. All experiments were performed in triplicate.

### 2.9. Western Blot

Transfected cells were lysed using a lysis buffer (R0030, Solarbio, Beijing, China) containing protease inhibitors (P6730, Solarbio, Beijing, China) to extract total protein, with concentrations determined by BCA assay. Protein samples were mixed with SDS-PAGE loading buffer (CW0027, CWBIO, Beijing, China), denatured at 100 °C for 10 min, separated via SDS-PAGE electrophoresis, and subsequently transferred to PVDF membranes using the wet transfer method. Membranes were blocked with 5% skim milk powder at room temperature for 2 h, washed with TBST, and then sequentially incubated with primary antibodies (overnight at 4 °C) and HRP-conjugated secondary antibodies (1 h at room temperature). Signals were developed using an ECL chemiluminescence system, and band intensity was quantified using ImageJ software (V2.0.0). Antibody details are provided in Appendix A.

### 2.10. Transcriptome Sequencing and Analysis of M. bovis -Infected BoMac Cells

Total RNA from *M. bovis*-infected BoMac cells (MOI = 1000, 24 h) and control cells (three biological replicates per group) was subjected to quality verification using an Agilent 2100 Bioanalyzer (2100 bioanalyzer, Agilent Technologies, Inc., Santa Clara, CA, USA). mRNA was enriched using Oligo (dT) magnetic beads, fragmented, and double-stranded cDNA was synthesized. Sequencing libraries were constructed through end repair, adenylation, adapter ligation, size selection (370–420 bp), and PCR amplification. Libraries were quantified using Qubit 2.0, and insert size distribution was validated with an Agilent 2100. Effective concentrations (>1.5 nM) were confirmed by qRT-PCR. High-throughput sequencing was performed by Novogene Bioinformatics Technology Co., Ltd. (Beijing, China) on an Illumina platform. Differentially expressed genes (DEGs) were identified using DESeq2 (v1.20.0) with an adjusted *p*-value (FDR) < 0.05. Functional enrichment analysis of Gene Ontology (GO) terms and KEGG pathways was conducted using Gene Set Enrichment Analysis (GSEA).

### 2.11. Construction and Transfection of Gadd45 and XIAP Gene Overexpression Plasmids and siRNAs

Based on bovine Gadd45α (GenBank: NC_037330.1) and XIAP (GenBank: NC_037357.1) gene sequences, signal-peptide-truncated coding sequences were synthesized and cloned into the pcDNA3.1 vector (Invitrogen, CA, USA) to generate C-terminal Myc-tagged fusion constructs. The resultant recombinant plasmids, designated pcDNA3.1-Gadd45α and pcDNA3.1-XIAP, were purified using the TIANpure Endotoxin-Free Plasmid Kit (Tiangen, Beijing, China).

### 2.12. Three siRNA Fragments Targeting Gadd45 mRNA (GenBank: NM_001034247.1) and XIAP mRNA (GenBank: NM_001205592.1) Were Designed and Synthesized by Sangon Biotech (China)

For Transfection, BoMac Cells (Seeded in 6-Well Plates at 70–80% Confluency) were Transfected with: 2.5 μg recombinant plasmid or Equivalent amounts of siRNA (e.g., 250 pmol/well) using Lipo8000 transfection reagent (Beyotime, Beijing, China). Transfection complexes were prepared by mixing plasmids/siRNA with 4 μL Lipo8000 in serum-free medium. After 24 h incubation (37 °C, 5% CO_2_), cells were washed three times with PBS and replenished with fresh complete medium. siRNA sequences are listed in Appendix A.

### 2.13. Quantitative Detection of Intracellular M. bovis

Following 24 h infection with *M. bovis*, culture supernatants from all experimental groups were aspirated, and cells were washed three times with PBS. Cells were lysed using 0.1% Triton X-100 (20107ES20, Yeasen, Shanghai, China) to release intracellular bacteria. The lysates were subjected to ten-fold serial dilutions in PBS, plated onto PPLO agar plates, and incubated at 37 °C under 5% CO_2_ for 7–10 days. Viable intracellular bacteria were quantified by enumerating colony-forming units (CFU).

### 2.14. Quantitative and Statistical Analysis

All data presented herein represent the results from three separate experiments and are mean ± SD. GraphPad Prism software (V8.3.1) was used to draw the figures in this paper. SPSS software (V21) was used for data statistical analysis. Immunoblotting results were semi-quantified using ImageJ software (V2.0.0). Data were compared with different groups using one-way analysis of variance (ANOVA), Student–Newman–Keuls (SNK) tests, and Student’s *t*-tests. ns, not significant (*p* > 0.05), 0.01 < ** p* <0.05, *** p* < 0.01, **** p* < 0.001, and ***** p* < 0.0001.

## 3. Results

### 3.1. Isolation, Identification and Characterization of M. bovis Xinjiang Isolate XJ01

This study systematically collected 84 nasal swabs and 4 tissue samples from eight cattle herds in Xinjiang (Figure 1A, Appendix A). Primary screening via UvrC-specific PCR (238 bp product, Appendix A) identified six *M. bovis*-positive isolates, further confirmed by UvrC and 16S rRNA gene amplification (Appendix A). 16S rRNA sequencing demonstrated that all six isolates belonged to the same strain, exhibiting classic “fried-egg” colonies on solid media (Appendix A). Biochemical characterization revealed that the isolate reduced triphenyltetrazolium chloride but did not hydrolyze gelatin, arginine, or esculin, decompose urea, ferment mannose, or utilize lactose/glucose (Appendix A). Antimicrobial susceptibility testing indicated sensitivity to nitrofurantoin, tetracyclines (tetracycline/doxycycline), and aminoglycosides (gentamicin/kanamycin), intermediate susceptibility to norfloxacin, and resistance to ciprofloxacin (Appendix A). Based on these molecular and phenotypic profiles, the strain was definitively identified as *M. bovis* and designated XJ01. Growth kinetics in PPLO broth showed a logarithmic phase beginning at 12 h, with peak titer (1.0 × 10^9^ CCU/mL) reached at 54 h, followed by a stationary phase maintained until 84 h (Appendix A), establishing 54-h cultures as optimal for infection experiments. Collectively, this work reports the first successful isolation and systematic characterization of *M. bovis* strain XJ01 in Xinjiang, providing a regionally prevalent strain with defined biological properties and antibiotic resistance patterns as a critical resource for local disease control and pathogenesis research.

### 3.2. Genomic Features of M. bovis Strain XJ01 Reveal High Coding Density and Metabolic Diversity

The genome of strain XJ01 comprises 1,023,741 bp with a GC content of 29.33%, encoding 939 predicted genes that occupy 88.48% of the genome. Gene length distribution peaks in the 200–1000 bp and >2000 bp ranges (Figure 1B). Functional annotation demonstrated that 34.5% of genes were classified by the COG database, dominated by ribosome biogenesis and translation-related genes (36.4%) (Figure 1C). GO terms annotated 65.9% of genes, with biological processes enriched in metabolic (417 genes) and cellular processes (388 genes), while molecular functions emphasized catalytic activity (379 genes) and binding capacity (324 genes) (Figure 1D). KEGG annotation covered 91.9% of genes, where metabolic pathways (31.7% of annotated genes) primarily constituted global metabolic maps (Figure 1E). Circos visualization further revealed non-uniform genomic distribution of coding genes, non-coding RNAs, and GC content/skew (blue/red indicating lower/higher GC content; green/orange denoting G < C or G > C skew) (Figure 2A). Collectively, the high coding density, predominance of translational machinery genes, and extensive metabolic functional annotations provide a molecular framework for investigating the ecological resilience and pathogenic mechanisms of XJ01.

### 3.3. Comparative Genomics and Evolutionary Analysis of M. bovis Strain XJ01

Whole-genome synteny analysis (Figure 2B) revealed near-complete structural conservation (>99%) between XJ01 and the prevalent Chinese strain HB0801, whereas translocations/inversions were observed relative to the reference strain PG45, confirming high conservation of core genomic architecture among regional isolates. Gene family clustering (Figure 2C) identified 587 conserved genes across nine analyzed strains, with XJ01 exhibiting a significantly expanded repertoire of strain-specific genes (n = 93). In contrast, other Chinese isolates including HB0801 (*n* = 3), NM2012 (n = 4), and Tibet-10 (*n* = 8) displayed minimal unique gene content, indicating low genetic divergence within domestic lineages. Phylogenetic reconstruction (Figure 2D) further demonstrated that XJ01 and HB0801 constitute the most recent evolutionary clade, while 08M, NM2012, Tibet-10, and Ningxia-1 formed a compact cluster, highlighting strong phylogeographic homogeneity among Chinese strains. Collectively, the conserved genome structure, distribution of accessory genes, and phylogenetic positioning establish XJ01 as a representative Chinese epidemic strain, providing genomic evidence for regional evolutionary divergence of *M. bovis*.

### 3.4. Establishment of an Apoptosis Model in BoMac Cells Induced by M. bovis Strain XJ01

To optimize infection conditions, apoptosis rates and cell viability were concurrently assessed using flow cytometry and CCK-8 assays. Time-course experiments (MOI = 1000) demonstrated a significant reduction in apoptosis at 24 h post-infection (32.4 ± 3.98%, *p* < 0.001 vs. PBS control 7.0 ± 1.80%), with parallel CCK-8 data confirming decreased viability (65.5 ± 1.98%, *p* < 0.01 vs. 0 h baseline 89.6 ± 1.02%). By 48 h, viability further deteriorated to 40.8 ± 1.27%, establishing 24 h as the critical window for apoptosis induction (Figure 3A–C). MOI-gradient assays (fixed 24 h duration) revealed maximal apoptosis at MOI = 1000 (35.4 ± 4.23%, *p* < 0.001 vs. uninfected control 10.9 ± 2.10%), accompanied by viability suppression to 53.1 ± 2.37% (*p* < 0.001 vs. control 87.8 ± 2.20%) (Figure 3D–F). Collectively, the parameters of MOI = 1000 at 24 h generate a robust apoptosis model (apoptosis > 35%, viability inhibition > 45%), providing a standardized infection system for subsequent mechanistic studies.

### 3.5. M. bovis XJ01 Induces Bax/Bcl-2-Mediated Apoptosis in BoMac Cells

To delineate the molecular mechanism underpinning *M. bovis* XJ01-induced apoptosis, spatiotemporal dynamics of the mitochondrial apoptosis regulators Bax and Bcl-2 were analyzed. At 24 h post-infection, qPCR demonstrated marked upregulation of pro-apoptotic Bax mRNA (*p* < 0.001 vs. PBS control) concomitant with significant downregulation of anti-apoptotic Bcl-2 (*p* < 0.05), resulting in a reduced Bcl-2/Bax ratio (*p* < 0.05) (Figure 4A). This regulatory pattern was amplified at the protein level: immunoblotting revealed increased Bax expression (*p* < 0.001), decreased Bcl-2 (*p* < 0.001), and a drastic decline in the Bcl-2/Bax ratio (*p* < 0.001) (Figure 4B,C). Collectively, XJ01 chronically disrupts Bax/Bcl-2 homeostasis in a time-dependent manner (peaking at 24 h), thereby activating the mitochondrial apoptotic pathway as the primary molecular driver of programmed macrophage death.

### 3.6. M. bovis XJ01 Modulates Apoptosis-Associated Gene Networks in Infected BoMac Cells

To delineate the host–pathogen interplay, transcriptomic profiling of *M. bovis* XJ01-infected versus control BoMac cells identified 9926 DEGs, comprising 1199 upregulated and 3224 downregulated transcripts (Figure 5A,B). GO enrichment analysis (adjusted *p* < 0.05) revealed significant overrepresentation of DEGs in biological processes including “nitrogen compound transport” and “cellular localization”, alongside molecular functions such as “RNA binding” and “catalytic activity acting on RNA” (Figure 5C). KEGG pathway analysis demonstrated broad enrichment in a range of pathways. While some enrichments (e.g., in neurodegeneration-associated pathways) likely reflect shared gene sets with core cellular stress responses rather than direct biological relevance, a focused annotation unequivocally identified the apoptosis and p53 signaling pathways as the most critically involved hubs (comparing Figure 5D and 5E). Integrative analysis prioritized 12 apoptosis-associated hub genes: Gadd45, BID, RAF, BAK, ENDOG, ARTS, CASP3, FAP1, FAS, ATM, Calpain, and XIAP. qRT-PCR validation confirmed pronounced upregulation of Gadd45 (pro-apoptotic) and downregulation of XIAP (anti-apoptotic regulator), consistent with RNA-seq trends (Figure 5F). Collectively, *M. bovis* XJ01 extensively reprograms the BoMac transcriptome, converging on RNA metabolism, substance transport, and apoptosis/p53 pathways, with Gadd45 and XIAP as core effectors modulating host cell death cascades.

### 3.7. M. bovis Targets Gadd45 and XIAP to Orchestrate Apoptotic Cascades in BoMac Cells

To define functional roles of Gadd45 and XIAP during infection, gene-specific perturbations were performed. Validated siRNA constructs (siGadd45-65 and siXIAP-1269) significantly reduced target mRNA levels (Appendix A). BoMac cells transfected with either pcDNA3.1-Gadd45 (overexpression) or siXIAP-1269 (knockdown) for 48 h were infected with *M. bovis* XJ01. Both Gadd45 overexpression and XIAP silencing markedly elevated pro-apoptotic Bax mRNA/protein while suppressing anti-apoptotic Bcl-2 expression (Figure 6A–D). Critically, these manipulations increased apoptosis rates by ~7.6% and 10% respectively (Figure 6E), confirming their pro-apoptotic potency.

Further assessment of pathogen fitness revealed that pro-apoptotic groups (siXIAP-1269 and pcDNA3.1-Gadd45α) exhibited significantly reduced intracellular bacterial loads at 12/24 h post-infection (*p* < 0.01 vs. infection-only control). Conversely, anti-apoptotic manipulations (siGadd45-65 and pcDNA3.1-XIAP) enhanced bacterial replication by 24 h (*p* < 0.05) (Figure 6F). Collectively, *M. bovis* XJ01 exploits Gadd45 upregulation and XIAP suppression to perturb the Bcl-2/Bax equilibrium, thereby triggering host apoptosis to curtail intramacrophage pathogen survival.

## 4. Discussion

*M. bovis* poses a substantial threat to bovine health due to its unique biological characteristics (e.g., lack of cell wall), frequently causing complex chronic BRD. Characteristic pathological manifestations include pneumonic lesions [20,21], infiltration of immune cells into tissues, and bronchial epithelial desquamation [22]. These attributes contribute to its resistance to multiple conventional antibiotics, significantly complicating clinical management and disease control [23]. Notably, the bovine respiratory mucosa harbors diverse mycoplasmas, with some species acting as pathogens (e.g., inducing bronchopneumonia) while others constitute commensal microbiota [24]. Focusing on Xinjiang region, this study prospectively collected 84 samples (nasal swabs, splenic and pulmonary tissues) from cattle with suspected infections across eight cities. UvrC-specific primers enabled precise differentiation of *M. bovis* from confounding species (*M. agalactiae* and *M. mycoides*), yielding a novel isolate phylogenetically clustered with the HB0801 strain, designated as XJ01. Whole-genome sequencing and phylogenetic analyses further established XJ01 as an emerging epidemic strain. This approach is critical for understanding contemporary epidemic dynamics, as demonstrated by recent phylodynamic studies that are unraveling the global spread and evolution of *M. bovis* [25].

*M. bovis* is a significant bovine pathogen that elicits pneumonia, arthritis, and mastitis through complex mechanisms including host cell apoptosis induction. Prevailing studies demonstrate its exploitation of apoptotic pathways for survival advantage: e.g., subverting autophagy via caspase-mediated Beclin 1 cleavage [17], triggering ER stress through P48 surface protein [26], secreting effector MbovP280 to antagonize anti-apoptotic CRYAB [27], upregulating CHOP to mediate ER-apoptosis crosstalk [28], or activating mitochondrial apoptosis via ROS burst and PTEN/PI3K-Akt-mTOR dysregulation [29,30]. Crucially, however, these mechanisms uniformly converge on pro-infection outcomes by facilitating immune evasion. This established paradigm highlights that M. bovis typically employs apoptosis as a precise weapon, deploying specific virulence factors (P48, MbovP280) to manipulate defined host pathways (ER stress, CRYAB inhibition) for its benefit. In stark contrast, our investigation of the Xinjiang epidemic strain XJ01 reveals a counterintuitive apoptosis phenotype: infection-triggered apoptosis constitutes a self-limiting host defense mechanism rather than a pathogen survival strategy. The key distinction lies not merely in the outcome, but in the underlying intent and execution. Whereas the classical mechanisms represent a pathogen-orchestrated sabotage of host signaling, the XJ01-induced response appears to be a host-directed elimination program that the pathogen fails to subvert. This paradigm shift is evidenced by: markedly reduced intracellular bacterial loads in pro-apoptotic groups (Gadd45 overexpression/XIAP knockdown), and enhanced pathogen replication when apoptosis is suppressed (XIAP overexpression/Gadd45 knockdown). We therefore hypothesize that this “pro-apoptosis/anti-pathogen” dichotomy may stem from the 93 XJ01-specific genes (Figure 2C), particularly those potentially involved in metabolic adaptation and virulence attenuation. This genomic feature could suggest an evolutionary selection for moderate immune activation to maintain infection homeostasis, a notion that requires functional validation in future studies. It is plausible that the absence or modification of potent anti-apoptotic effectors (akin to MbovP280) in XJ01, or the expression of novel antigens that potently trigger the Gadd45/XIAP axis, underpins this altered host–pathogen dynamic. While targeted enhancement of this pathway (e.g., XIAP silencing) suggests a potential therapeutic strategy in vitro by curtailing bacterial survival, our data also provide a note of caution: any such approach would require extremely precise regulation, as excessive apoptosis may exacerbate bronchial epithelial desquamation and pulmonary tissue damage—hallmarks of advanced BRD pathology.

Cell apoptosis, governed by the dynamic equilibrium between pro-apoptotic Bax and anti-apoptotic Bcl-2, serves as a core rheostat for tissue homeostasis. Our study demonstrates that *M. bovis* XJ01 infection in BoMac cells activates the p53 signaling pathway, markedly upregulating pro-apoptotic Gadd45 while suppressing XIAP expression. This bidirectional dysregulation disrupts the Bcl-2/Bax balance, culminating in mitochondrial apoptotic cascade initiation (Figure 7). This finding extends the canonical apoptosis paradigm: although elevated Bcl-2/Bax ratio as an apoptotic hallmark exhibits cross-species conservation—validated in vertebrate models (hepatic fibrosis [31], environmental toxin-induced grass carp hepatotoxicity [32], human thyrocyte dysfunction [33]) and invertebrate systems (marine rotifer stress response [34])—our work delineates its pathogen-triggered molecular axis. Furthermore, our transcriptomic data revealed significant modulation of the MAPK signaling pathway and autophagy, suggesting a complex interplay between XJ01-induced apoptosis and other immune processes such as cytokine production and phagosome maturation. The precise role of these related immune factors, including reactive oxygen species (ROS) generation, in conjunction with the apoptotic response, remains an important and fascinating topic for future investigation. Specifically, we identify: Gadd45/XIAP dual-control node as the critical effector coupling p53 activation to mitochondrial apoptosis during mycoplasmal infection.

Our transcriptomic profiling identified DEGs intricately linked to apoptotic pathways, with qRT-PCR validation confirming consistent expression trends. Among these, Gadd45 (significantly upregulated) and XIAP (markedly downregulated) emerged as the most prominent regulators. The GADD45 protein family (including GADD45α isoforms) and XIAP serve as critical stress-response mediators governing cell cycle control, DNA repair, apoptotic commitment, and tumorigenesis [35,36,37]. Notably, both factors exhibit vital functions in viral infections: GADD45 restricts HIV-1 replication by suppressing viral transcription [38] and negatively regulates JNK signaling through MKK7 targeting [39], whereas XIAP modulates Ripoptosome-mediated cell death via degradative inhibition [37]. Crucially, however, the functional atlas of the GADD45/XIAP axis remains uncharacterized in bacterial pathogens, particularly wall-less *M. bovis* infections. This study unravels that *M. bovis* XJ01 infection in macrophages specifically upregulates GADD45 while suppressing XIAP, thereby disequilibrating the BCL-2/BAX balance and activating the mitochondrial apoptosis pathway. This regulatory paradigm contrasts sharply with GADD45’s transcription-repressive role in antiviral defense [38] and XIAP’s canonical anti-apoptotic function in oncology [37]—in the context of XJ01 infection, their coordinated pro-apoptotic drive unexpectedly serves not as a conduit for pathogen immune evasion but as an important host defense mechanism for intracellular bacterial clearance. We hypothesize that GADD45 upregulation exerts antibacterial effects through dual plausible mechanisms: partly through mimicking its JNK-inhibitory property [39] to block mycoplasma-induced hyperinflammation, while concurrently amplifying p53-mediated apoptotic signaling [36] to directly eliminate infected cells. Concomitantly, XIAP downregulation liberates caspase activity from its inhibition [37], accelerating programmed eradication of parasitized host cells.

Several limitations of this study should be considered. First, the CFU assay, despite including rigorous washing steps, may not completely distinguish between internalized and tightly adherent extracellular bacteria. However, as this potential confounder is present across all experimental groups, the significant reduction in CFU observed in the pro-apoptotic groups likely represents a robust and conservative estimate of the true reduction in intracellular bacterial load. Second, the high MOI (1000) used was necessary to elicit a robust apoptotic response for mechanistic dissection within our experimental timeframe and may not precisely reflect physiological conditions during the initial phase of infection. Furthermore, while our data show the apoptotic response persists at 48 h, longer-term kinetics under lower infectious doses remain to be fully characterized. Future studies using lower MOIs over extended periods or in vivo models will be crucial to understand the kinetics and full pathological relevance of this apoptotic pathway.

Translating these findings to in vivo conditions, we postulate that the efficacy of this Gadd45/XIAP-mediated apoptotic defense could be a critical determinant of infection outcome. An early and robust activation may facilitate bacterial clearance and resolve acute infection, whereas a delayed or weak response might permit bacterial persistence, contributing to the chronicity and tissue damage characteristic of advanced *M. bovis* disease. This paradigm not only provides a mechanistic explanation for the spectrum of clinical manifestations but also suggests that modulating the Gadd45/XIAP axis could be explored as a potential host-directed therapeutic strategy in future studies, to determine if it could aid clearance in refractory cases.

## 5. Conclusions

This study successfully isolated the *M. bovis* Xinjiang epidemic strain XJ01, and phylogenetic analysis demonstrated its closest evolutionary relationship to the HB0801 strain. Through the establishment of a BoMac cell infection model combined with transcriptomic profiling, we observed significant enrichment of differentially expressed genes in the p53 signaling and apoptosis pathways, leading to the identification of key regulators GADD45 and XIAP. Functional validation experiments confirmed that GADD45 overexpression and XIAP silencing synergistically increased apoptotic rates while significantly reducing intracellular *M. bovis* survival. The integrated findings reveal a novel host defense mechanism: XJ01 infection induces host-protective apoptosis by coordinately upregulating GADD45 and downregulating XIAP, thereby disrupting the BCL-2/BAX balance and activating the mitochondrial apoptotic cascade (Figure 7). This process represents a self-limiting antimicrobial strategy where apoptotic elimination of infected cells functions to limit bacterial replication.

## Figures and Tables

**Figure 1 microorganisms-13-02031-f001:**
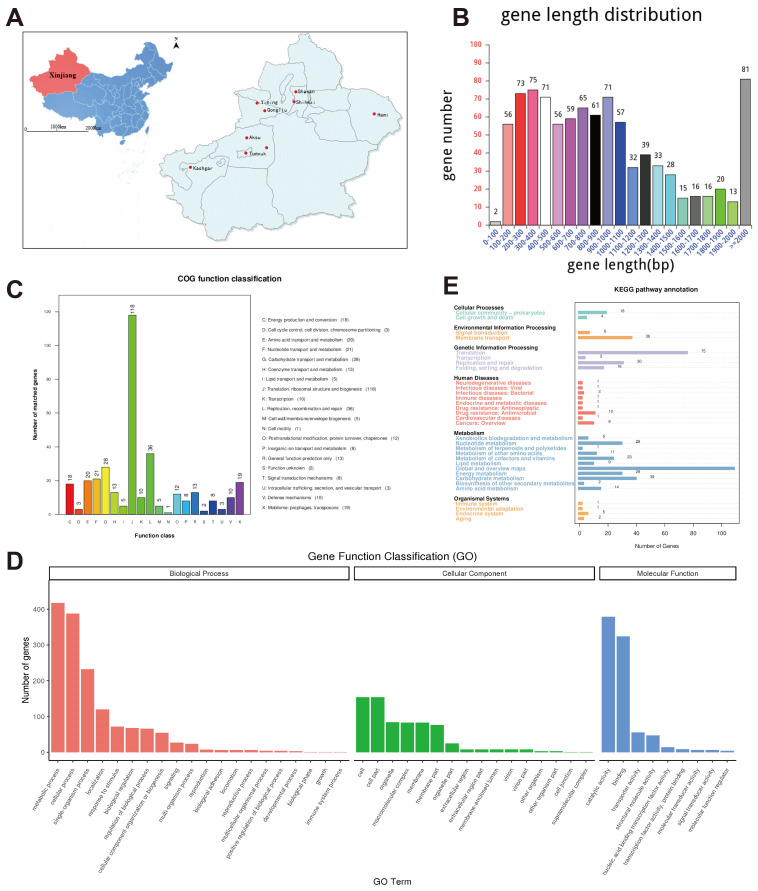
Sampling distribution and genomic signatures of *Mycoplasma bovis* (*M. bovis*) strain XJ01. (**A**) Spatial distribution of sampling sites across Xinjiang (8 cattle herds; 84 nasal swabs plus 4 tissue samples). (**B**) Protein-coding gene length distribution of XJ01 strain: 939 genes predominantly clustered in 200–1000 bp (67.2%) and >2000 bp (29.8%) ranges. (**C**) COG functional classification: Translation/ribosome biogenesis constituted the largest category (118/324, 36.4%) among 34.5% annotated genes. (**D**) GO term annotation: Metabolic processes (417/619) and catalytic activity (379/619) represented dominant biological functions. (**E**) KEGG pathway distribution: Metabolism-associated genes accounted for 31.7% of annotations (112/274 mapped to global metabolic pathways).

**Figure 2 microorganisms-13-02031-f002:**
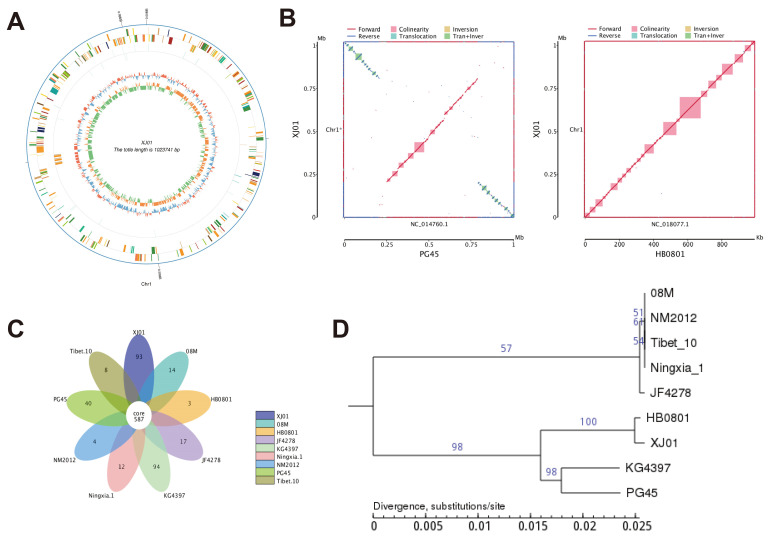
Genomic architecture and phylogenetic relationships of *M. bovis* strain XJ01. (**A**) Circular genome map: Illustrates the 1,023,741 bp circular genome (GC content 29.33%). Concentric rings from outer to inner: genome coordinates, protein-coding genes (orange), non-coding RNAs (green), GC content (blue < mean/red > mean), GC skew (green: G < C/orange: G > C). (**B**) Collinearity analysis: Left panel reveals localized translocations/inversions between XJ01 and reference strain PG45; Right panel shows near-complete structural conservation (>99.9% synteny) with Chinese isolate HB0801. (**C**) Gene family analysis: 587 core genes conserved across nine strains; XJ01 harbors 93 unique genes (significantly exceeding HB0801’s 3 genes), indicating low genetic differentiation in Xinjiang isolates. (**D**) Maximum-likelihood phylogenetic tree: XJ01 clusters with HB0801 as nearest neighbors (bootstrap > 95%); Distinct evolutionary clade formed by 08M/NM2012/Tibet-10/Ningxia-1 strains demonstrates geographical adaptive divergence.

**Figure 3 microorganisms-13-02031-f003:**
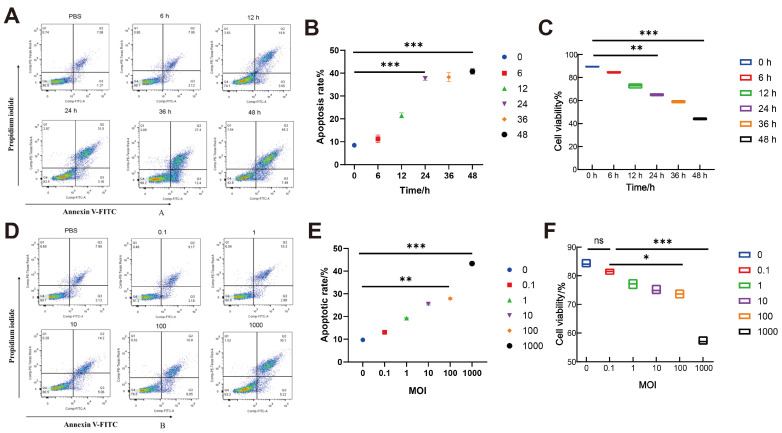
Optimization of *M. bovis* XJ01 infection conditions in BoMac cells. (**A**,**B**) Time-course apoptosis analysis (MOI = 1000): Representative flow cytometry plots (**A**) and quantification (**B**) demonstrate significantly increased apoptotic rate at 24 h (32.4 ± 3.98%; *p* < 0.05 vs. 12 h: 26.91 ± 3.23%). (**C**) Time-dependent viability assay: CCK-8 confirms severe viability suppression at 24 h (65.5 ± 1.98%; *p* < 0.001 vs. 0 h: 89.6 ± 1.02%). (**D**,**E**) MOI-gradient apoptosis analysis (24 h): Flow cytometry profiles (**D**) and statistical analysis (**E**) reveal maximal apoptosis induction at MOI = 1000 (35.4 ± 4.23%; *p* < 0.01 vs. MOI = 0.1). (**F**) MOI-dependent viability assay: CCK-8 concurrently shows minimal viability at MOI = 1000 (53.1 ± 2.37%; *p* < 0.001 vs. control: 87.8 ± 2.20%). All data presented herein represent the results from three separate experiments and are mean ± SD. ns, not significant (*p* > 0.05), 0.01 < * *p* < 0.05, ** *p* < 0.01 and *** *p* < 0.001.

**Figure 4 microorganisms-13-02031-f004:**
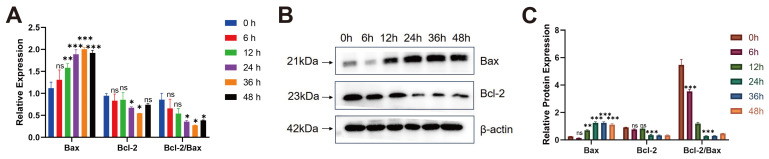
*M. bovis* XJ01 activates mitochondrial apoptosis via Bax/Bcl-2 axis dysregulation. (**A**) Bax/Bcl-2 mRNA dynamics: At 24 h post-infection, qPCR reveals highly significant Bax upregulation (*p* < 0.001 vs. PBS), concurrent Bcl-2 downregulation (*p* < 0.05), and reduced Bcl-2/Bax ratio (*p* < 0.05). (**B**) Representative immunoblotting: Time-dependent accumulation of pro-apoptotic Bax protein and attenuation of anti-apoptotic Bcl-2 (24 h peak effect). (**C**) Densitometric quantification: Bcl-2/Bax ratio decreased dramatically by 82% at 24 h (*p* < 0.001), validating mitochondrial apoptosis commitment. All data presented herein represent the results from three separate experiments and are mean ± SD. ns, not significant (*p* > 0.05), 0.01 < * *p* < 0.05, ** *p* < 0.01 and *** *p* < 0.001.

**Figure 5 microorganisms-13-02031-f005:**
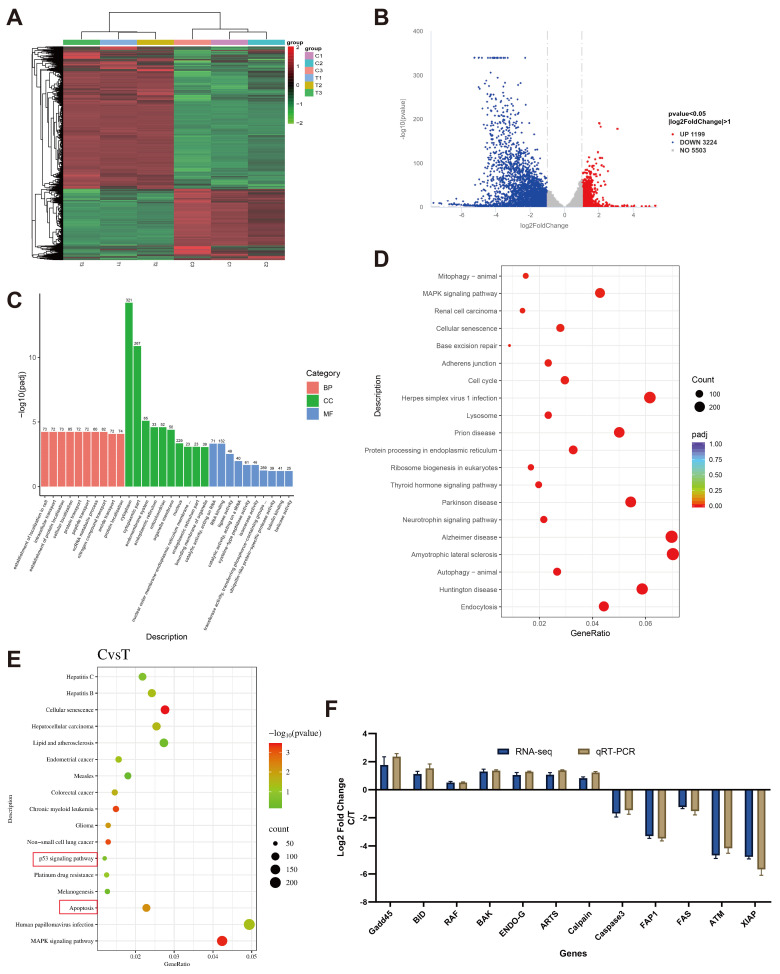
Transcriptomic profiling and pathway enrichment in *M. bovis* XJ01-infected BoMac cells. (**A**) Differentially expressed genes (DEGs) heatmap: Hierarchical clustering of 9926 DEGs (|log_2_ Fold Change | ≥ 1, padj ≤ 0.05) between infected (XJ01) and control (PBS) groups. Rows: genes; columns: biological replicates (*n* = 3). (**B**) Volcano plot of DEGs: Distribution of transcriptional alterations. Red: 1199 significantly upregulated genes (log_2_ Fold Change >1, padj < 0.05); Blue: 3224 downregulated genes (log_2_FC < −1, padj < 0.05). Dashed lines indicate significance thresholds. (**C**) Gene Ontology (GO) enrichment: Significantly enriched terms (*p* < 0.05) in Biological Process (BP), Cellular Component (CC), and Molecular Function (MF) categories. Top 10 terms per category displayed. (**D**) Kyoto Encyclopedia of Genes and Genomes (KEGG) pathway enrichment: Bubble plot of significantly perturbed pathways (padj < 0.05). Circle size: gene count; color intensity: enrichment significance. (**E**) Key signaling pathway enrichment: Scatter plot highlighting apoptosis and p53 pathways (padj < 0.001). Rich factor = DEGs annotated in pathway/total DEGs. (**F**) qRT-PCR validation of apoptosis-related genes: Expression changes (log_2_ Fold Change) of 12 candidate genes. Gadd45 and XIAP showed most pronounced alterations, concordant with RNA-seq data.

**Figure 6 microorganisms-13-02031-f006:**
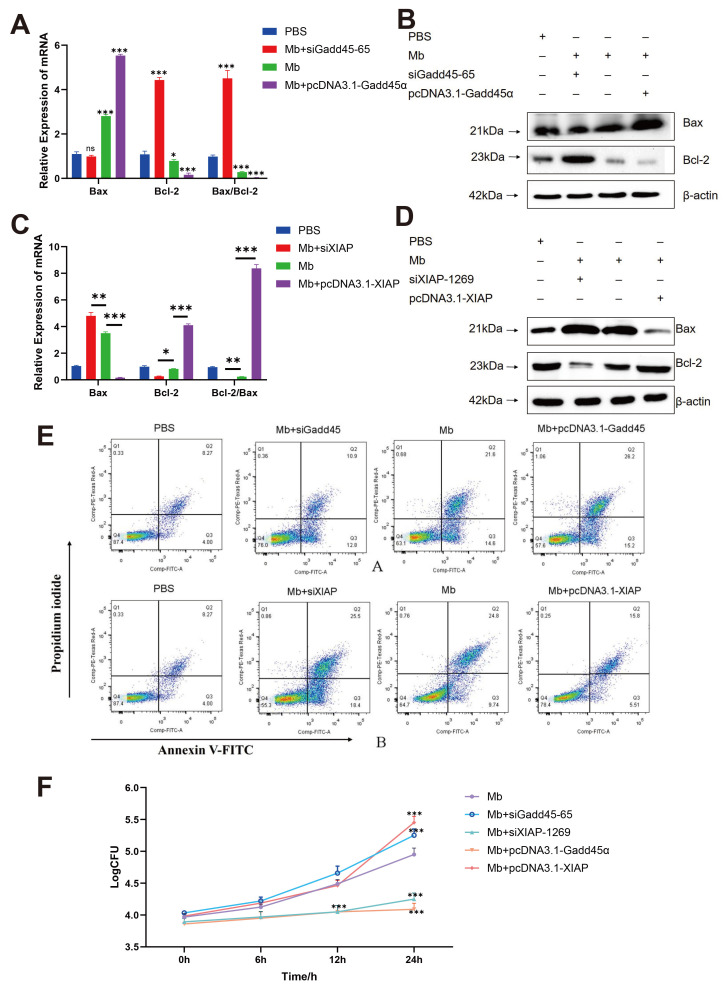
Functional validation of Gadd45 overexpression and XIAP knockdown on *M. bovis*-induced apoptosis and intracellular proliferation. (**A**,**C**) qRT-PCR analysis of apoptotic regulators: Bax and Bcl-2 mRNA levels in BoMac cells transfected with either pcDNA3.1-Gadd45α (Gadd45 OE), XIAP-targeting siRNA (siXIAP-1269), Gadd45-targeting siRNA (siGadd45-65), or pcDNA3.1-XIAP (XIAP OE) for 48 h prior to XJ01 infection. (**B**,**D**) Immunoblotting of apoptotic effectors: Corresponding protein expression profiles under identical treatments. β-actin served as loading control. (**E**) Flow cytometric quantification of apoptosis: Significant escalation of apoptotic cells in Gadd45 OE and siXIAP groups post-infection. (**F**) Intracellular bacterial burden dynamics: CFU assays revealed enhanced *M. bovis* replication in Gadd45 OE and siXIAP groups at 24 h, inversely correlating with host cell viability. All data presented herein represent the results from three separate experiments and are mean ± SD. ns, not significant (*p* > 0.05), 0.01 < * *p* < 0.05, ** *p* < 0.01 and *** *p* < 0.001.

**Figure 7 microorganisms-13-02031-f007:**
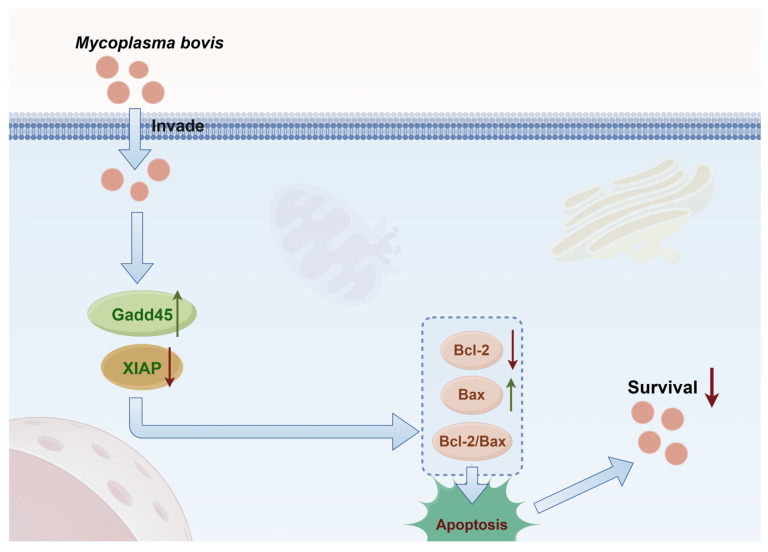
In vitro mechanistic model of *M. bovis* XJ01-induced apoptosis via the Gadd45/XIAP-Bax/Bcl-2 axis in BoMac cells. This integrative schematic, based on findings from our cell culture model, delineates the apoptotic pathway initiated by *M. bovis* XJ01 infection: Upon invading host cells, the pathogen activates cellular stress signaling, which concurrently upregulates pro-apoptotic Gadd45 expression and suppresses anti-apoptotic XIAP expression. This transcriptional reprogramming promotes the activation of pro-apoptotic effector Bax while inhibiting anti-apoptotic regulator Bcl-2, thereby disrupting the Bcl-2/Bax stoichiometric equilibrium. The resultant mitochondrial outer membrane permeabilization (MOMP) triggers the caspase cascade, ultimately executing apoptosis to facilitate intracellular pathogen clearance. (The proposed model requires further validation in vivo.) (Mechanistic diagram created with FigDraw 2.0). In the diagram, red downward arrows indicate inhibition (suppression of protein expression), and green upward arrows indicate promotion (activation or enhancement of protein expression).

**Table 1 microorganisms-13-02031-t001:** Reference strain.

Reference Strain Name	Reference Strain Genomic ID	Separated Area
PG45	NC_014760	MD, USA
HB0801	NC_018077	Hubei, China
Ningxia-1	NZ_CP023663	Ningxia, China
Tibet-10	NZ_CP062195	Qinghai, China
08M	CP019639.1	Gansu, China
NM2012	CP011348.1	Inner Mongolia, China
JF4278	LT578453.1	Bern, Switzerland
KG4397	AP019558	Tokyo, Japan

## Data Availability

The RNA-seq sequencing data generated in this study have been deposited in the NCBI Sequence Read Archive (SRA) under the accession number PRJNA1308958. The data are currently under embargo until 21 August 2029 (or upon publication of associated follow-up studies, whichever comes first), after which they will be freely available. All other data needed to evaluate the conclusions in the paper are present in the paper and/or the Appendix A.

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
