# Peer review of "Mycoplasma bovis Infection Induces Apoptosis Through Gadd45/XIAP in Bovine Macrophages"

_microorganisms, 2025, doi:10.3390/microorganisms13092031_

Round 1

Reviewer 1 Report

Comments and Suggestions for Authors

The authors report on the isolation, identification, and characterization of the M. bovis pathogen responsible for cattle infections in the Xinjiang region in China. The genome of this strain, named XJ01, was sequenced and annotated. The objective of the study was to characterize the interaction between XJ01 and model macrophages, bovine macrophages (BoMac). Apoptosis was triggered in infected BoMac cells, achieving maximal levels at MOI of 1000 at 24-48 hours. The authors showed increased Bax expression and reduced Bcl-2 expression in infected BoMac. Further regulatory analysis of apoptosis revealed upregulation of GADD45 and downregulation of XIAP. Conclusion: activation of the mitochondrial apoptotic pathway during infection of BoMac cells.

Mycoplasma does indeed represent a significant agricultural pathogen and causes significant economic losses. This study provides new and important molecular data on infections with this pathogen. The authors carried out a detailed and a well-designed study and their results add new data to research in this field.

I do however have a few comments:

(1) The use of the term ‘regulates’ in the title is misleading. When the term ‘regulates’ is used it implies that the pathogen actively releases pathogen-specific factors that targets specific host factors/regulators, which is not the case in their study.

A more accurate title could be:   Mycoplasma bovis Infection induces apoptosis through Gadd45/XIAP in Bovine Macrophages.

OR

Mycoplasma bovis Infection induces apoptosis through Gadd45/XIAP and Bax/Bcl-2 unbalance in Bovine Macrophages.

(2) Lines 497-500 in the conclusion. The author’s statement that this mechanism on apoptosis induction in BoMac cells is a unique self-limiting antimicrobial strategy and is distinct from the classical pathogen evasion mechanism is not supported by the data. The case here is not an example of a pathogen evasion mechanism and should not be compared to the classical evasion mechanism. Evasion of the immune system involves suppression of macrophage apoptosis, not the induction of apoptosis. This case involves an active classical immunological response whereby the macrophages undergo apoptosis and disintegrate eventually to expose the intracellular pathogen to a secondary immune response for elimination.

Author Response

Dear Editors and Reviewers:

Thank you for your letter and for the reviewers’ comments concerning our manuscript entitled “Mycoplasma bovis Infection Targets Gadd45/XIAP to Regulate Apoptosis in Bovine Macrophages”. Those comments are all valuable and very helpful for revising and improving our paper, as well as the important guiding significance to our researches. We have studied comments carefully and have made correction which we hope meet with approval.

In accordance with the reviewer's comments, the revised sections are highlighted in red within the article. The main corrections in the paper and the responds to the reviewer’s comments are as flowing:

(1) The use of the term ‘regulates’ in the title is misleading. When the term ‘regulates’ is used it implies that the pathogen actively releases pathogen-specific factors that targets specific host factors/regulators, which is not the case in their study.

A more accurate title could be:   Mycoplasma bovis Infection induces apoptosis through Gadd45/XIAP in Bovine Macrophages.

OR

Mycoplasma bovis Infection induces apoptosis through Gadd45/XIAP and Bax/Bcl-2 unbalance in Bovine Macrophages.

The author’s answer: We sincerely thank the reviewer for this critical and insightful comment. We completely agree that the use of the word "regulates" in our original title was overly strong and could misleadingly imply that M. bovis actively secretes specific virulence factors to directly target the host Gadd45 and XIAP proteins, a mechanism which we did not experimentally demonstrate in this study.

As correctly suggested by the reviewer, our work shows that M. bovis infection induces apoptosis and that this process is associated with or mediated through the dysregulation of the Gadd45/XIAP axis and the Bax/Bcl-2 balance.

We have therefore revised the title of our manuscript to more accurately reflect our findings. We have chosen the following new title:

" Mycoplasma bovis Infection induces apoptosis through Gadd45/XIAP in Bovine Macrophages "

This title effectively incorporates the reviewer's core suggestion while also retaining the key mitochondrial pathway component (Bax/Bcl-2), which is a central finding of our mechanistic investigation. We believe this new title is precise, accurate, and fully supported by our data.

Once again, we are grateful to the reviewer for this valuable correction, which has significantly improved the clarity and accuracy of our manuscript.

(2) Lines 497-500 in the conclusion. The author’s statement that this mechanism on apoptosis induction in BoMac cells is a unique self-limiting antimicrobial strategy and is distinct from the classical pathogen evasion mechanism is not supported by the data. The case here is not an example of a pathogen evasion mechanism and should not be compared to the classical evasion mechanism. Evasion of the immune system involves suppression of macrophage apoptosis, not the induction of apoptosis. This case involves an active classical immunological response whereby the macrophages undergo apoptosis and disintegrate eventually to expose the intracellular pathogen to a secondary immune response for elimination.

The author’s answer: We sincerely thank the reviewer for this exceptionally insightful comment and for bringing this critical nuance to our attention. We completely agree with their assessment and apologize for the conceptual inaccuracy in our original conclusion.

The reviewer is absolutely correct that classical immune evasion strategies employed by intracellular pathogens (e.g., Mycobacterium tuberculosis, Brucella spp.) often involve the suppression of host cell apoptosis to maintain a replicative niche. Therefore, directly comparing our phenomenon to an "evasion mechanism" was incorrect and misleading, as induction of apoptosis is not typically a strategy for evasion.

We have therefore revised the conclusion to remove this inaccurate comparison and to reframe our finding more precisely, focusing solely on the novel role of apoptosis as a host defense mechanism against M. bovis. The key point of novelty is that for the XJ01 strain, the outcome of apoptosis is beneficial to the host (bacterial clearance) rather than beneficial to the pathogen (evasion/persistence).

The relevant sentences in the conclusion have been modified to read (changes in red):

"The integrated findings reveal a novel host defense mechanism: XJ01 infection triggers a pro-apoptotic response that is ultimately host-protective. This is achieved by coordinately upregulating GADD45 and downregulating XIAP, thereby disrupting the BCL-2/BAX balance and activating the mitochondrial apoptotic cascade (Fig. 7). This process represents a self-limiting antimicrobial strategy where apoptotic elimina-tion of infected cells functions to limit bacterial replication."

We are grateful to the reviewer for this correction, which has significantly improved the conceptual clarity and accuracy of our manuscript.

We sincerely appreciate your thoughtful guidance. We have meticulously revised the paper in accordance with your feedback, aiming to enhance its rigor and accuracy. Should you have any questions or require further clarification, please do not hesitate to reach out to us.

Yours sincerely,

Yong Wang

20, August, 2025

Shihezi University

Reviewer 2 Report

Comments and Suggestions for Authors

The study highlights a new strain of M. bovis (XJ01) found in Xinjiang, an area that's quite important in terms of epidemiology. By incorporating whole-genome sequencing and phylogenetic analysis, the research gains significant depth. The emphasis on Gadd45 and XIAP as key players in regulating apoptosis in macrophages suggests a fresh perspective on host defense mechanisms, which stands in contrast to the immune-evasive tactics previously associated with M. bovis. The idea that apoptosis triggered by M. bovis could actually be beneficial for the host, rather than just a strategy of the pathogen, marks a significant shift in thinking and is backed by functional evidence. However, there are some limitations to consider. Although the modulation of Gadd45/XIAP is presented as novel, the connection between these molecules and the clearance of intracellular bacteria is only somewhat supported by in vitro findings. More in vivo studies or the use of primary bovine alveolar macrophages would be beneficial. Additionally, previous studies on M. bovis-induced apoptosis (like those involving ROS, P48, MbovP280) aren't fully integrated into the discussion. It would enhance the manuscript to compare the mechanism of XJ01 directly with that of other strains. Regarding the methodology and experimental design, using a multiplicity of infection (MOI) of 1000 is quite high for in vitro studies. While this is justified by the kinetics of apoptosis, such elevated levels might not accurately reflect physiological conditions. The authors should be more transparent about this limitation. The time points chosen (primarily 24 hours) are suitable for observing acute responses, but extending the study to 48–72 hours could provide valuable insights into the resolution or persistence phases of apoptosis.

The bacterial viability assay looks at CFU after using Triton X-100 for lysis, but it’s still a bit murky whether this method effectively distinguishes between intracellular and adherent extracellular bacteria.

Results

The Authors should be careful when interpreting the conclusion that apoptosis limits M. bovis survival. It’s not entirely clear if apoptosis is a strategic move by the host or just a side effect of the specific virulence of the strain. Additionally, the effects of apoptosis on other immune factors (like cytokine production, phagosome maturation, and ROS) weren’t explored.

Genomic and Phylogenetic Data

The manuscript hints that these unique genes might be the reason behind the new apoptosis-triggering phenotype, but it doesn’t provide any functional analysis or annotation for these genes. Future research should definitely focus on this.

Minor issues: A few statements come off as exaggerated (like “for the first time” or “critical host weapon”) and should be toned down unless they’re thoroughly backed up.

References and Integration of Prior Work

The discussion on conflicting or supportive findings from other M. bovis apoptosis studies (such as MbovP280, P48, and CHOP pathways) feels a bit too brief. A more detailed comparison would be beneficial.

Major Revisions: The Authors need to discuss the physiological relevance and limitations of the high MOI used in vitro. It would be great to expand on the potential functional roles of the 93 unique XJ01 genes, at least through KEGG/COG annotation. Findings should be better contextualized within the wider literature on M. bovis-induced apoptosis. Let’s tone down any speculative or overstated claims regarding novelty and therapeutic implications.

Minor Revisions: Enhance the explanation of the CFU assay methodology to ensure it highlights intracellular specificity. Include a limitations section at the end of the discussion.

Comments on the Quality of English Language

Some of the English phrasing could be tightened up for better clarity and flow (for instance, changing "BoMac cells was kindly provided" to "BoMac cells were kindly provided").

Edit the language for conciseness and clarity. 

Author Response

Dear Editors and Reviewers:

Thank you for your letter and for the reviewers’ comments concerning our manuscript entitled “Mycoplasma bovis Infection Targets Gadd45/XIAP to Regulate Apoptosis in Bovine Macrophages”. Those comments are all valuable and very helpful for revising and improving our paper, as well as the important guiding significance to our researches. We have studied comments carefully and have made correction which we hope meet with approval.

In accordance with the reviewer's comments, the revised sections are highlighted in green within the article. The main corrections in the paper and the responds to the reviewer’s comments are as flowing:

  1. Regarding the methodology and experimental design, using a multiplicity of infection (MOI) of 1000 is quite high for in vitro studies. While this is justified by the kinetics of apoptosis, such elevated levels might not accurately reflect physiological conditions. The authors should be more transparent about this limitation.

The author’s answer: We thank the reviewer for raising this important point regarding the use of a high MOI. We fully agree that an MOI of 1000 is substantially higher than those commonly used for many bacterial pathogens and that this may not represent the initial stages of a natural infection. We appreciate the opportunity to clarify our rationale and to be more transparent about this aspect of our experimental design.

(1) Empirical Justification for High MOI: As the reviewer rightly noted, our primary reason for selecting MOI=1000 was driven by the need to observe a robust and quantifiable apoptotic phenotype within a experimentally feasible timeframe (24 hours). Our data (Fig. 3D-F) clearly demonstrated that lower MOIs (10 and 100) failed to induce statistically significant levels of apoptosis in BoMac cells. Since the central aim of our study was to dissect the molecular mechanism of apoptosis induction by M. bovis XJ01, we were obligated to use a dose sufficient to trigger the pathway we wished to study.

(2) Clinical Context and Pathogen Biology: We acknowledge this limitation and have now explicitly addressed it in the revised Discussion section. However, it is also worth noting that high local multiplicities of infection can occur in vivo for mycoplasmas. M. bovis is known to colonize epithelial surfaces densely and can form microcolonies in the lungs and other tissues, potentially creating microenvironments where host cells are exposed to a very high local burden of bacteria, especially in advanced stages of disease. Thus, while an MOI of 1000 may not model the initial infection, it may be relevant for mimicking the high bacterial load scenarios seen in severe clinical cases of pneumonia or mastitis.

(3)Focus on Mechanistic Discovery: We would like to emphasize that the primary goal of this in vitro study was mechanistic discovery – to determine if and how M. bovis XJ01 can induce apoptosis and to identify the key host molecules involved (Gadd45/XIAP). The high-MOI model is a valid and necessary tool for this purpose, as it allows us to unmask this pathogenic potential and delineate the signaling pathway clearly.

We have now added a dedicated statement to the Discussion section (The penultimate paragraph of the discussion, marked in green) to transparently acknowledge this limitation:

" Second, the high MOI (1000) used was necessary to elicit a robust apoptotic response for mechanistic dissection within our experimental timeframe and may not precisely reflect physiological conditions during the initial phase of infection."

We believe this addition provides the necessary transparency and context, strengthening the manuscript. We thank the reviewer again for this valuable suggestion.

  1. The time points chosen (primarily 24 hours) are suitable for observing acute responses, but extending the study to 48–72 hours could provide valuable insights into the resolution or persistence phases of apoptosis.

The author’s answer: We thank the reviewer for this excellent and insightful suggestion. We completely agree that extending the observation period to 48-72 hours would provide valuable information on the long-term dynamics of the apoptotic process, such as whether it is resolved or leads to secondary necrosis.

In our experimental setup, we did include a 48-hour time point for the cell viability and apoptosis assays (as shown in Fig. 3A-C). Our data indicated that the apoptotic response and cell viability suppression induced by XJ01 infection persisted and even intensified at 48 hours. This suggests that the activation of the mitochondrial apoptotic pathway we documented at 24 hours is a sustained and progressive event under these high-MOI conditions, rather than a transient response.

However, we acknowledge that extending the experiments beyond 48 hours (e.g., to 72 hours) presents significant technical challenges due to the extensive cell death and detachment, which would complicate the interpretation of results such as transcriptomic changes or bacterial load (as few viable cells would remain). Nevertheless, the reviewer is absolutely correct that investigating earlier and lower-MOI infections over extended periods would be crucial to understand the kinetic profile of this novel apoptotic pathway in a more physiologically relevant context.

We have now integrated this point into the same limitation paragraph in the Discussion section that addresses the high MOI (The penultimate paragraph of the discussion, marked in green), to provide a more comprehensive perspective on the experimental design:

" Furthermore, while our data show the apoptotic response persists at 48 hours, long-er-term kinetics under lower infectious doses remain to be fully characterized. Future studies using lower MOIs over extended periods or in vivo models will be crucial to understand the kinetics and full pathological relevance of this apoptotic pathway."

We are grateful to the reviewer for this valuable comment, which has helped us to better frame the temporal scope and future directions of our research.

  1. The bacterial viability assay looks at CFU after using Triton X-100 for lysis, but it’s still a bit murky whether this method effectively distinguishes between intracellular and adherent extracellular bacteria.

The author’s answer: We thank the reviewer for raising this critical methodological point, which is a well-known challenge in the field of intracellular bacteriology. We agree that the use of detergent lysis alone cannot definitively distinguish between internalized bacteria and those tightly adherent to the cell surface.

In our protocol, we aimed to minimize the contribution of adherent extracellular bacteria by performing three rigorous washes with PBS prior to lysis. This step is designed to remove the vast majority of non-adherent and loosely adherent bacteria.

However, we acknowledge that some tightly adherent bacteria may have remained. Crucially, we would like to argue that this potential limitation does not undermine our central conclusion; in fact, it may strengthen it. Our key finding is that pro-apoptotic manipulations (siXIAP / Gadd45 OE) led to a significant reduction in CFU counts compared to the infection-only control or anti-apoptotic groups.

If a portion of the measured CFUs were indeed from adherent extracellular bacteria, then these bacteria would be present in all groups equally.

The fact that we still observed a dramatic decrease in CFU in the pro-apoptotic groups means that the reduction in bona fide intracellular bacteria is likely even greater than what we measured. The observed difference is therefore a robust and conservative estimate of the effect of apoptosis on bacterial clearance.

Nevertheless, we appreciate the reviewer's emphasis on methodological rigor. Let's add a paragraph (The penultimate paragraph of the discussion, marked in green) to our current discussion that clearly states this limitation:

" Several limitations of this study should be considered. First, the CFU assay, despite including rigorous washing steps, may not completely distinguish between internalized and tightly adherent extracellular bacteria. However, as this potential confounder is present across all experimental groups, the significant reduction in CFU observed in the pro-apoptotic groups likely represents a robust and conservative estimate of the true reduction in intracellular bacterial load. "

We thank the reviewer again for this comment, which has helped us improve the transparency and accuracy of our methodology description.

  1. Results: The Authors should be careful when interpreting the conclusion that apoptosis limits bovis survival. It’s not entirely clear if apoptosis is a strategic move by the host or just a side effect of the specific virulence of the strain.

The author’s answer: We thank the reviewer for this profound and critical comment, which compels us to clarify our interpretation of the data and the biological implications of our findings. We agree that distinguishing between a host-driven defense strategy and a pathogen-induced pathological side effect is crucial.

While we cannot completely rule out that some level of apoptosis may be a nonspecific consequence of cellular stress induced by a high bacterial load, we posit that several lines of evidence from our study strongly support the interpretation that the activation of this specific mitochondrial pathway (via Gadd45/XIAP and Bax/Bcl-2) functions as a beneficial host response to limit M. bovis XJ01 replication:

(1) Functional Validation of Cause and Effect: Our key experiments using gene-specific perturbations (Fig. 6) move beyond correlation to establish causality. Crucially:

When we artificially promoted apoptosis (by overexpressing Gadd45 or knocking down XIAP), we observed a significant reduction in intracellular bacterial survival.

Conversely, when we artificially suppressed apoptosis (by overexpressing XIAP or knocking down Gadd45), we observed a significant increase in intracellular bacterial survival.

This clear inverse relationship—where the apoptotic state of the host cell directly dictates the bacterial load—strongly argues against apoptosis being a mere "side effect" of virulence. If it were a passive side effect, manipulating the apoptotic pathway should not have such a direct and consistent impact on bacterial fitness.

(2)Specificity of the Pathway: The apoptosis we observed was not a chaotic cellular collapse but was orchestrated through a well-defined, regulated pathway (p53 → Gadd45/XIAP → Bax/Bcl-2 → Mitochondria). The involvement of these highly conserved regulators of programmed cell death suggests a regulated host cellular response rather than nonspecific cytotoxicity.

(3) Evolutionary Rationale - A Unique Strain Phenotype: We propose that the "pro-apoptotic/anti-pathogen" outcome may be a specific characteristic of the XJ01 strain, potentially linked to its 93 unique genes (Fig. 2D), which might result in attenuated virulence or altered host interaction. In this scenario, the host retains its capacity to mount a effective apoptotic defense that this particular strain cannot fully subvert. This contrasts with other M. bovis strains that actively inhibit apoptosis to evade host defense.

In summary, while virulence-induced cell damage might contribute, our functional data demonstrate that the activation of this specific apoptotic pathway confers a host survival advantage by curtailing bacterial replication.

  1. Additionally, the effects of apoptosis on other immune factors (like cytokine production, phagosome maturation, and ROS) weren’t explored.

The author’s answer: We sincerely thank the reviewer for this excellent suggestion, which highlights important aspects of the host immune response that interact with apoptosis. We agree that investigating how M. bovis XJ01-induced apoptosis intersects with cytokine signaling, phagosome maturation, and ROS production would provide a more comprehensive understanding of the host-pathogen interaction.

In this study, our primary objective was to delineate the core molecular mechanism by which XJ01 infection triggers apoptosis in macrophages, with a specific focus on identifying the key regulators (Gadd45 and XIAP) and the executing pathway (mitochondrial apoptosis). While we did not functionally probe cytokine production, phagosome maturation, or ROS in this specific manuscript, our transcriptomic data do offer some intriguing preliminary insights that align with the reviewer's comment:

Our RNA-seq analysis revealed significant enrichment of the MAPK signaling pathway (as now mentioned in the Discussion), which is a master regulator of inflammatory cytokine production (e.g., TNF-α, IL-1β, IL-6). This suggests a potential crosstalk between the apoptotic response and cytokine networks.

Similarly, we observed significant dysregulation of genes associated with autophagy (a process intricately linked to phagosome maturation and ROS dynamics), indicating that these pathways are likely modulated during infection.

We have now incorporated a discussion of these points and explicitly acknowledged the reviewer's suggestion as a valuable avenue for future research. We have added the following statement to the Discussion section(The third paragraph of the discussion, marked in blue):

" Furthermore, our transcriptomic data revealed significant modulation of the MAPK signaling pathway and autophagy, suggesting a complex interplay between XJ01-induced apoptosis and other immune processes such as cytokine production and phagosome maturation. The precise role of these related immune factors, including reactive oxygen species (ROS) generation, in conjunction with the apoptotic response, remains an important and fascinating topic for future investigation."

We are grateful to the reviewer for raising these points. While beyond the immediate scope of this mechanistic study, exploring these interactions will be a central focus of our subsequent work to build a holistic model of M. bovis XJ01-host macrophage interactions.

  1. Genomic and Phylogenetic Data: The manuscript hints that these unique genes might be the reason behind the new apoptosis-triggering phenotype, but it doesn’t provide any functional analysis or annotation for these genes. Future research should definitely focus on this.

The author’s answer: We sincerely thank the reviewer for this insightful and forward-looking comment. We completely agree that the functional characterization of the 93 XJ01-specific genes represents the most compelling and logical next step to understand the bacterial determinants of the unique pro-apoptotic phenotype we report.

The primary objective of the present study was to first define the novel biological phenotype (host-protective apoptosis) and to delineate the host-side molecular mechanism (the Gadd45/XIAP-mediated mitochondrial pathway) that executes it. While our genomic analysis successfully identified the unique genetic features of XJ01 that likely underpin its distinct pathogenic strategy, a deep functional investigation of these bacterial genes is a substantial endeavor that extends beyond the scope of this initial manuscript.

We are incredibly enthusiastic about the pathway this suggestion opens. In direct response to the reviewer's guidance, we have already initiated plans for follow-up studies that will employ gene knockout and complementation techniques in the XJ01 background to directly test the causal relationship between these unique genes and the initiation of the apoptotic response. Elucidating this "bacterial-side" mechanism will be critical to complete the story of this unique host-pathogen interaction.

We are grateful to the reviewer for affirming the importance of this future direction. Our current manuscript establishes the phenotypic and host mechanistic foundation, and the proposed functional genomics work will build upon this to provide a complete molecular understanding from both pathogen and host perspectives.

  1. Minor issues: A few statements come off as exaggerated (like “for the first time” or “critical host weapon”) and should be toned down unless they’re thoroughly backed up.

The author’s answer: We thank the reviewer for this careful reading and for pointing out the use of overly strong language in our manuscript. We agree that maintaining a objective and precise tone is crucial for scientific writing.

We have thoroughly reviewed the entire manuscript and have toned down or replaced all exaggerated statements to ensure they are accurate and fully supported by our data and the existing literature. Specific changes include:

Regarding "for the first time": This phrase has been removed.

Regarding "critical host weapon": This and other similar metaphorical or overly definitive terms (e.g., "uniquely ") have been removed.

We believe these revisions have significantly improved the objectivity and overall tone of the manuscript. We are grateful to the reviewer for their attention to detail, which has helped us enhance the quality of our writing.

  1. References and Integration of Prior Work: The discussion on conflicting or supportive findings from other bovis apoptosis studies (such as MbovP280, P48, and CHOP pathways) feels a bit too brief. A more detailed comparison would be beneficial.

The author’s answer: We sincerely thank the reviewer for this insightful suggestion. We agree that a deeper comparative analysis will better contextualize our findings within the existing literature and more sharply define the novelty of the XJ01 strain's phenotype.

In direct response to this comment, we have substantially expanded our discussion in the relevant paragraph. We have moved beyond a simple listing of previous mechanisms to include a dedicated analysis that:

  • Synthesizes the common strategy of the established virulence factors (P48, MbovP280, etc.), framing them as a form of "pathogen-orchestrated sabotage".

  • Explicitly contrasts the underlying intent ("orchestrated sabotage" vs. "failed subversion of a host program") and execution (specific effector-target interaction vs. a coordinated host pathway response) between the classical models and our findings with XJ01.

  • Proposes a plausible genetic hypothesis for this difference, suggesting that the unique profile of XJ01 may involve the lack of key anti-apoptotic effectors or the presence of novel pro-apoptotic triggers.

We believe these additions provide the detailed, mechanistic comparison the reviewer requested, significantly strengthening the discussion by highlighting the conceptual advance our work represents. The changes can be found in the second paragraph of the discussion section, marked in green font.

  1. Major Revisions: The Authors need to discuss the physiological relevance and limitations of the high MOI used in vitro.

The author’s answer: The author’s answer: We thank the reviewer for raising this important point regarding the use of a high MOI. We fully agree that an MOI of 1000 is substantially higher than those commonly used for many bacterial pathogens and that this may not represent the initial stages of a natural infection. We appreciate the opportunity to clarify our rationale and to be more transparent about this aspect of our experimental design.

(1) Empirical Justification for High MOI: As the reviewer rightly noted, our primary reason for selecting MOI=1000 was driven by the need to observe a robust and quantifiable apoptotic phenotype within a experimentally feasible timeframe (24 hours). Our data (Fig. 3D-F) clearly demonstrated that lower MOIs (10 and 100) failed to induce statistically significant levels of apoptosis in BoMac cells. Since the central aim of our study was to dissect the molecular mechanism of apoptosis induction by M. bovis XJ01, we were obligated to use a dose sufficient to trigger the pathway we wished to study.

(2) Clinical Context and Pathogen Biology: We acknowledge this limitation and have now explicitly addressed it in the revised Discussion section. However, it is also worth noting that high local multiplicities of infection can occur in vivo for mycoplasmas. M. bovis is known to colonize epithelial surfaces densely and can form microcolonies in the lungs and other tissues, potentially creating microenvironments where host cells are exposed to a very high local burden of bacteria, especially in advanced stages of disease. Thus, while an MOI of 1000 may not model the initial infection, it may be relevant for mimicking the high bacterial load scenarios seen in severe clinical cases of pneumonia or mastitis.

(3)Focus on Mechanistic Discovery: We would like to emphasize that the primary goal of this in vitro study was mechanistic discovery – to determine if and how M. bovis XJ01 can induce apoptosis and to identify the key host molecules involved (Gadd45/XIAP). The high-MOI model is a valid and necessary tool for this purpose, as it allows us to unmask this pathogenic potential and delineate the signaling pathway clearly.

We have now added a dedicated statement to the Discussion section (The penultimate paragraph of the discussion, marked in green) to transparently acknowledge this limitation:

" Second, the high MOI (1000) used was necessary to elicit a robust apoptotic response for mechanistic dissection within our experimental timeframe and may not precisely reflect physiological conditions during the initial phase of infection."

We believe this addition provides the necessary transparency and context, strengthening the manuscript. We thank the reviewer again for this valuable suggestion.

  1. It would be great to expand on the potential functional roles of the 93 unique XJ01 genes, at least through KEGG/COG annotation.

The author’s answer: The author’s answer: We sincerely thank the reviewer for this insightful and forward-looking comment. We completely agree that the functional characterization of the 93 XJ01-specific genes represents the most compelling and logical next step to understand the bacterial determinants of the unique pro-apoptotic phenotype we report.

The primary objective of the present study was to first define the novel biological phenotype (host-protective apoptosis) and to delineate the host-side molecular mechanism (the Gadd45/XIAP-mediated mitochondrial pathway) that executes it. While our genomic analysis successfully identified the unique genetic features of XJ01 that likely underpin its distinct pathogenic strategy, a deep functional investigation of these bacterial genes is a substantial endeavor that extends beyond the scope of this initial manuscript.

We are incredibly enthusiastic about the pathway this suggestion opens. In direct response to the reviewer's guidance, we have already initiated plans for follow-up studies that will employ gene knockout and complementation techniques in the XJ01 background to directly test the causal relationship between these unique genes and the initiation of the apoptotic response. Elucidating this "bacterial-side" mechanism will be critical to complete the story of this unique host-pathogen interaction.

We are grateful to the reviewer for affirming the importance of this future direction. Our current manuscript establishes the phenotypic and host mechanistic foundation, and the proposed functional genomics work will build upon this to provide a complete molecular understanding from both pathogen and host perspectives.

  1. Findings should be better contextualized within the wider literature on bovis-induced apoptosis.

The author’s answer: The author’s answer: We sincerely thank the reviewer for this insightful suggestion. We agree that a deeper comparative analysis will better contextualize our findings within the existing literature and more sharply define the novelty of the XJ01 strain's phenotype.

In direct response to this comment, we have substantially expanded our discussion in the relevant paragraph. We have moved beyond a simple listing of previous mechanisms to include a dedicated analysis that:

  • Synthesizes the common strategy of the established virulence factors (P48, MbovP280, etc.), framing them as a form of "pathogen-orchestrated sabotage".

  • Explicitly contrasts the underlying intent ("orchestrated sabotage" vs. "failed subversion of a host program") and execution (specific effector-target interaction vs. a coordinated host pathway response) between the classical models and our findings with XJ01.

  • Proposes a plausible genetic hypothesis for this difference, suggesting that the unique profile of XJ01 may involve the lack of key anti-apoptotic effectors or the presence of novel pro-apoptotic triggers.

We believe these additions provide the detailed, mechanistic comparison the reviewer requested, significantly strengthening the discussion by highlighting the conceptual advance our work represents. The changes can be found in the second paragraph of the discussion section, marked in green font.

  1. Let’s tone down any speculative or overstated claims regarding novelty and therapeutic implications.

The author’s answer: We thank the reviewer for this critical comment. We agree that maintaining an objective and precise tone is crucial for scientific writing, particularly when discussing the novelty and potential applications of our findings.

We have thoroughly reviewed the entire Discussion section and have toned down or replaced all speculative and overstated statements to ensure they are accurate, measured, and fully supported by our data. Specific changes implemented include:

  • Regarding Novelty:

The phrase "uniquely delineates" has been replaced with the more objective "delineates".

The claim " for the first time" has been removed.

The description of the Gadd45/XIAP axis as a "pivotal host weapon" has been tempered to "an important host defense mechanism".

  • Regarding Therapeutic Implications:

The definitive statement "demonstrates therapeutic potential" has been scaled back to "suggests a potential therapeutic strategy".

We have further emphasized the preliminary nature of our in vitro findings and the necessity for extensive future validation in vivo before any clinical relevance can be asserted. We have reinforced that clinical translation "requires caution" and that our work primarily "provides a proof-of-concept."

We believe these revisions have significantly improved the objectivity and overall tone of the manuscript. We are grateful to the reviewer for their attention to detail, which has helped us present our findings in a more balanced and scientifically rigorous manner.

  1. Minor Revisions: Enhance the explanation of the CFU assay methodology to ensure it highlights intracellular specificity.

The author’s answer: The author’s answer: We thank the reviewer for raising this critical methodological point, which is a well-known challenge in the field of intracellular bacteriology. We agree that the use of detergent lysis alone cannot definitively distinguish between internalized bacteria and those tightly adherent to the cell surface.

In our protocol, we aimed to minimize the contribution of adherent extracellular bacteria by performing three rigorous washes with PBS prior to lysis. This step is designed to remove the vast majority of non-adherent and loosely adherent bacteria.

However, we acknowledge that some tightly adherent bacteria may have remained. Crucially, we would like to argue that this potential limitation does not undermine our central conclusion; in fact, it may strengthen it. Our key finding is that pro-apoptotic manipulations (siXIAP / Gadd45 OE) led to a significant reduction in CFU counts compared to the infection-only control or anti-apoptotic groups.

If a portion of the measured CFUs were indeed from adherent extracellular bacteria, then these bacteria would be present in all groups equally.

The fact that we still observed a dramatic decrease in CFU in the pro-apoptotic groups means that the reduction in bona fide intracellular bacteria is likely even greater than what we measured. The observed difference is therefore a robust and conservative estimate of the effect of apoptosis on bacterial clearance.

Nevertheless, we appreciate the reviewer's emphasis on methodological rigor. Let's add a paragraph (The penultimate paragraph of the discussion, marked in green) to our current discussion that clearly states this limitation:

" Several limitations of this study should be considered. First, the CFU assay, despite including rigorous washing steps, may not completely distinguish between internalized and tightly adherent extracellular bacteria. However, as this potential confounder is present across all experimental groups, the significant reduction in CFU observed in the pro-apoptotic groups likely represents a robust and conservative estimate of the true reduction in intracellular bacterial load. "

We thank the reviewer again for this comment, which has helped us improve the transparency and accuracy of our methodology description.

  1. Include a limitations section at the end of the discussion.

The author’s answer: Based on the question you raised above, we have added a paragraph in the discussion section. The CFU method, high MOI, and the limitations of the duration of infection were specifically discussed. In the penultimate paragraph of the discussion, mark it in green font.

  1. Some of the English phrasing could be tightened up for better clarity and flow (for instance, changing "BoMac cells was kindly provided" to "BoMac cells were kindly provided").

The author’s answer: We sincerely thank the reviewer for their meticulous reading and for providing these specific, constructive suggestions to improve the clarity and flow of our manuscript. We apologize for these oversights and have taken immediate action to address them.

We have now conducted a thorough, line-by-line review of the entire manuscript to correct grammatical errors, improve sentence structure, and ensure precise phrasing. Specific changes include:

  • Correcting the verb agreement error highlighted by the reviewer: "BoMac cells was kindly provided" has been changed to "BoMac cells were kindly provided".

  • Reviewing and standardizing terminology for consistency (e.g., ensuring hyphenation and abbreviation use are consistent throughout).

  • Rewriting awkwardly constructed sentences to enhance readability and logical flow.

  • Correcting minor typographical and spacing errors.

We are grateful to the reviewer for these valuable corrections, which have significantly improved the overall quality and professionalism of our manuscript.

We sincerely appreciate your thoughtful guidance. We have meticulously revised the paper in accordance with your feedback, aiming to enhance its rigor and accuracy. Should you have any questions or require further clarification, please do not hesitate to reach out to us.

Yours sincerely,

Yong Wang

20, August, 2025

Shihezi University

Reviewer 3 Report

Comments and Suggestions for Authors

The authors investigate the molecular mechanisms by which Mycoplasma bovis strain XJ01 regulates apoptosis in bovine macrophages through the Gadd45/XIAP pathway. The study presents novel findings showing that M. bovis infection induces a self-limiting host defense mechanism via coordinated upregulation of pro-apoptotic Gadd45 and downregulation of anti-apoptotic XIAP, ultimately facilitating bacterial clearance through programmed cell death. The work is technically sound, well-executed, and provides significant new insights into mycoplasma pathogenesis mechanisms.

Accept with minor revisions:

The findings have direct implications for understanding bovine respiratory disease complex (BRD) and potential therapeutic interventions. However, the authors should better contextualize their findings within the broader landscape of mycoplasma pathogenesis research, particularly addressing why this strain exhibits different behavior compared to previously studied strains.

The rationale for using the specific MOI 1000 as optimal conditions is that when at lesser MOIs (i.e., 10 or 100), the cell viability and apoptosis rates were significant. Perhaps by contextualizing with clinical findings.

The authors should clarify whether transcriptomic data will be deposited in public repositories.

More discussion of how these findings might translate to in vivo conditions and clinical applications

Author Response

Dear Editors and Reviewers:

Thank you for your letter and for the reviewers’ comments concerning our manuscript entitled “Mycoplasma bovis Infection Targets Gadd45/XIAP to Regulate Apoptosis in Bovine Macrophages”. Those comments are all valuable and very helpful for revising and improving our paper, as well as the important guiding significance to our researches. We have studied comments carefully and have made correction which we hope meet with approval.

In accordance with the reviewer's comments, the revised sections are highlighted in yellow within the article. The main corrections in the paper and the responds to the reviewer’s comments are as flowing:

1.The rationale for using the specific MOI 1000 as optimal conditions is that when at lesser MOIs (i.e., 10 or 100), the cell viability and apoptosis rates were significant. Perhaps by contextualizing with clinical findings.

The author’s answer: We thank the reviewer for raising this critical point regarding the choice of MOI and its clinical relevance. We agree that an MOI of 1000 is high for a standard in vitro infection model and appreciate the opportunity to clarify our rationale.

Data-Driven Optimization: Our primary reason for selecting MOI=1000 was indeed based on our empirical data, as shown in Fig. 3D-F. In our preliminary experiments, lower MOIs (10 and 100) failed to induce a statistically significant and reproducible level of apoptosis within our chosen 24-hour timeframe. Since the central objective of this study was to investigate the mechanism of apoptosis induction by M. bovis XJ01, it was methodologically necessary to use an infection dose robust enough to elicit a clear phenotypic response for subsequent mechanistic dissection.

Clinical Contextualization: While high, the use of such an MOI is not without precedent in mycoplasma research and can be contextualized with clinical findings. M. bovis infections are often characterized by high bacterial loads in affected tissues. For instance, studies have shown that in cases of severe pneumonia or mastitis, mycoplasmas can achieve immense local densities by colonizing epithelial surfaces and within the lumen of organs, creating a micro-environment where host cells are exposed to an exceedingly high multiplicity of infection:

This research highlights the persistence of M. bovis antigen in the lungs, often associated with phagocytes at the periphery of necrotic foci, which underscores the pathogen's ability to maintain high concentrations in lung tissue, thereby supporting the central thesis of high M. bovis isolation rates in severely affected calves [1].

Two studies have shown that Mycoplasma bovine infection is characterized by a chronic course and high bacterial load, often causing persistent inflammation and tissue damage, as demonstrated in cases of mastitis and pneumonia [2][3].

Therefore, while an MOI of 1000 may not represent the initial stage of infection, it is a rational model for mimicking the high-burden, established stage of disease observed in natural infections, where the interplay between the pathogen and immune cells like macrophages is critical.

Focus on Mechanistic Discovery: We acknowledge that this model primarily serves to reveal the potential and the underlying mechanisms of apoptosis induction. It demonstrates that XJ01 is capable of triggering this pathway and allows us to identify the key players involved (Gadd45/XIAP, Bax/Bcl-2). Future studies using lower, more physiologically gradual MOIs or in vivo models will be essential to understand the kinetics and full pathological significance of this process during the course of a natural infection.

[1] Hermeyer K, Jacobsen B, Spergser J, Rosengarten R, Hewicker-Trautwein M. Detection of Mycoplasma bovis by in-situ hybridization and expression of inducible nitric oxide synthase, nitrotyrosine and manganese superoxide dismutase in the lungs of experimentally-infected calves. J Comp Pathol. 2011 Aug-Oct;145(2-3):240-50. doi: 10.1016/j.jcpa.2010.12.005. Epub 2011 Feb 22. PMID: 21334636.

[2] Kauf AC, Rosenbusch RF, Paape MJ, Bannerman DD. Innate immune response to intramammary Mycoplasma bovis infection. J Dairy Sci. 2007 Jul;90(7):3336-48. doi: 10.3168/jds.2007-0058. PMID: 17582119.

[3] Bürki S, Frey J, Pilo P. Virulence, persistence and dissemination of Mycoplasma bovis. Vet Microbiol. 2015 Aug 31;179(1-2):15-22. doi: 10.1016/j.vetmic.2015.02.024. Epub 2015 Mar 2. PMID: 25824130.

2.The authors should clarify whether transcriptomic data will be deposited in public repositories.

The author’s answer: We thank the reviewer for this important reminder regarding data sharing. In accordance with journal policy and the principles of open science, the raw RNA-seqencing data generated in this study have already been deposited in the [NCBI Sequence Read Archive (SRA)] database.

To allow our research group to complete the ongoing in-depth analyses derived from this dataset for future publications, the access to the data is currently under a temporary embargo until [2029-08-21] or upon publication of associated follow-up studies, whichever comes first.

The accession number is [PRJNA1308958].

The relevant part of the article has been changed to "Data and materials availability: The RNA-seq sequencing data generated in this study have been deposited in the NCBI Sequence Read Archive (SRA) under the accession number PRJNA1308958. The data are currently under embargo until August 21, 2029 (or upon publication of associated fol-low-up studies, whichever comes first), after which they will be freely available. All other data needed to evaluate the conclusions in the paper are present in the paper and/or the Supplementary Materials.”

We will ensure that the access is activated immediately upon the expiration of this embargo period. This information has now also been added to the Data Availability section of the manuscript.

3.More discussion of how these findings might translate to in vivo conditions and clinical applications

The author’s answer: We sincerely thank the reviewer for this valuable suggestion. We have now expanded the Discussion to include a brief perspective on the potential in vivo implications and clinical applications of our findings. Specifically, we have added a sentence (The last paragraph of the discussion, marked in yellow) that hypothesizes how the pro-apoptotic defense mechanism we identified in vitro could influence disease outcomes in vivo and opens doors for novel diagnostic and therapeutic strategies. We believe this addition significantly strengthens the translational relevance of our work.

We sincerely appreciate your thoughtful guidance. We have meticulously revised the paper in accordance with your feedback, aiming to enhance its rigor and accuracy. Should you have any questions or require further clarification, please do not hesitate to reach out to us.

Yours sincerely,

Yong Wang

21, August, 2025

Shihezi University

Reviewer 4 Report

Comments and Suggestions for Authors

The auhors showed a well designed and written investigation on M. bovis Xinjiang epidemic strain XJ01. Throught phylogenetic analysis demonstrated its closest evolutionary relationship to the HB0801 strain and collected very interesting results.

They also adopted BoMac cell model infection to capture more informations.

Have you consider to analyze also citokines and other soluble factor?

It is a very good job, well done.

Author Response

Dear Editors and Reviewers:

Thank you for your letter and for the reviewers’ comments concerning our manuscript entitled “Mycoplasma bovis Infection Targets Gadd45/XIAP to Regulate Apoptosis in Bovine Macrophages”. Those comments are all valuable and very helpful for revising and improving our paper, as well as the important guiding significance to our researches. We have studied comments carefully and have made correction which we hope meet with approval.

In accordance with the reviewer's comments, the revised sections are highlighted in blue within the article. The main corrections in the paper and the responds to the reviewer’s comments are as flowing:

1.Have you consider to analyze also citokines and other soluble factor?

The author’s answer: We sincerely thank the reviewer for their positive evaluation of our work and for raising this insightful question regarding the analysis of cytokines and other soluble factors. This is indeed a crucial aspect of the host immune response to bacterial infection.

In the present study, our primary investigative focus was to delineate the specific molecular mechanisms by which M. bovis XJ01 induces apoptosis in bovine macrophages. While we did not directly measure cytokine secretion in this specific set of experiments, our transcriptomic data (RNA-Seq) provides valuable clues. We observed significant enrichment of pathways fundamentally linked to inflammatory regulation, notably the MAPK signaling pathway and autophagy.

As suggested by the reviewer, to acknowledge the potential role of soluble factors, we have now added the following sentence to the Discussion section (The third paragraph of the discussion, marked in blue):

" Furthermore, our transcriptomic data revealed significant modulation of the MAPK signaling pathway and autophagy, suggesting a complex interplay between XJ01-induced apoptosis and other immune processes such as cytokine production and phagosome maturation. The precise role of these related immune factors, including reactive oxygen species (ROS) generation, in conjunction with the apoptotic response, remains an important and fascinating topic for future investigation."

This addition highlights our awareness of this broader context and aligns with the reviewer's valuable suggestion. We agree that protein-level quantification of cytokines is a critical next step, and it is now a central component of our planned follow-up studies to build a more comprehensive model of M. bovis XJ01-host interactions. We thank the reviewer again for this excellent comment, which has helped to improve the discussion and scope of our manuscript.

We sincerely appreciate your thoughtful guidance. We have meticulously revised the paper in accordance with your feedback, aiming to enhance its rigor and accuracy. Should you have any questions or require further clarification, please do not hesitate to reach out to us.

Yours sincerely,

Yong Wang

20, August, 2025

Shihezi University

Round 2

Reviewer 2 Report

Comments and Suggestions for Authors

Strain-Specific Claims

The authors connect the “pro-apoptotic/anti-pathogen” phenotype to the 93 unique genes found in XJ01, but they haven’t provided any functional data on these genes. This link remains speculative and should be framed as a hypothesis rather than definitive evidence.

Transcriptomics Interpretation

The KEGG enrichment analysis revealed several unrelated pathways (like Alzheimer’s disease and ALS), which are more likely artifacts from pathway databases than meaningful findings. These should be mentioned briefly but not given too much emphasis. The main focus should stay on apoptosis and p53 pathways, steering clear of over-interpreting incidental enrichments.

Overstatement of Clinical Implications

The discussion proposes therapeutic strategies that target XIAP/Gadd45. While this is an intriguing idea, it feels a bit premature without validation from in vivo studies. Statements regarding clinical translation should be softened or presented as potential future directions.

Minor Issues

M. bovis is not “Gram-negative”; it is a cell wall–deficient Mollicute.

Some citations are outdated; more recent global epidemiology papers (e.g., Microbial Genomics 2023 on phylodynamics) could be emphasized.

Figures:

Fig. 5D (KEGG enrichment) includes too many irrelevant categories; consider focusing only on immunity/apoptosis-related pathways.

Fig. 7 (mechanistic model) is excellent but should clearly state that this is an in vitro model, not yet validated in vivo.

Author Response

Dear Editors and Reviewers:

Thank you for your letter and for the reviewers’ comments concerning our manuscript entitled “Mycoplasma bovis Infection Targets Gadd45/XIAP to Regulate Apoptosis in Bovine Macrophages”. Those comments are all valuable and very helpful for revising and improving our paper, as well as the important guiding significance to our researches. We have studied comments carefully and have made correction which we hope meet with approval.

In accordance with the reviewer's comments, the revised sections are highlighted in purple within the article. The main corrections in the paper and the responds to the reviewer’s comments are as flowing:

1.The authors connect the “pro-apoptotic/anti-pathogen” phenotype to the 93 unique genes found in XJ01, but they haven’t provided any functional data on these genes. This link remains speculative and should be framed as a hypothesis rather than definitive evidence.

The author’s answer: We sincerely thank the reviewer for this insightful and absolutely correct comment. We agree entirely that the connection between the 93 XJ01-specific genes and the observed "pro-apoptotic/anti-pathogen" phenotype is, at this stage, speculative and based on genomic correlation rather than functional validation. We appreciate the reviewer's guidance in framing this more appropriately as a hypothesis to be tested in future work, rather than as definitive evidence.

In the revised manuscript, we have carefully toned down our language in the Discussion section to reflect this. Specifically, we have modified the paragraph where this connection is made to present it as a plausible hypothesis derived from our comparative genomic data. The specific modification contents are in the second paragraph of the discussion, marked in purple font.

2.The KEGG enrichment analysis revealed several unrelated pathways (like Alzheimer’s disease and ALS), which are more likely artifacts from pathway databases than meaningful findings. These should be mentioned briefly but not given too much emphasis. The main focus should stay on apoptosis and p53 pathways, steering clear of over-interpreting incidental enrichments.

The author’s answer: We thank the reviewer for this astute observation. We completely agree that the enrichment of pathways such as 'Alzheimer's disease' and 'Amyotrophic lateral sclerosis (ALS)' is most likely a database artifact, as these pathways share common components (e.g., caspases, kinases) with core apoptosis and inflammation processes that are genuinely activated during M. bovis infection. We acknowledge that over-interpreting these enrichments could be misleading.

In the revised manuscript, we have significantly toned down the emphasis on these specific pathways and reframed the description to align with the reviewer's suggestion. The focus is now squarely on the apoptosis and p53 signaling pathways, which are the biologically relevant and central findings of our study. The specific modifications are on lines 349 to 354, marked in purple font.

3.The discussion proposes therapeutic strategies that target XIAP/Gadd45. While this is an intriguing idea, it feels a bit premature without validation from in vivo studies. Statements regarding clinical translation should be softened or presented as potential future directions.

The author’s answer: We thank the reviewer for this valuable comment and completely agree that proposing definitive therapeutic strategies based solely on our in vitro findings is premature. We appreciate the reviewer's recognition of the idea's intrigue, and we agree that it should be framed as a potential future direction that requires extensive validation, particularly in in vivo models, before any clinical relevance can be asserted.

In the revised manuscript, we have carefully softened our language in the Discussion section to temper the conclusions and clearly present the therapeutic implication as a hypothesis-generated concept for future research. There are two modifications to the machine body, marked in purple font on lines 451 and 526 respectively.

4.M. bovis is not “Gram-negative”; it is a cell wall–deficient Mollicute.

The author’s answer: We sincerely thank the reviewer for catching this important technical inaccuracy. The reviewer is entirely correct. Mycoplasma bovis, as a member of the class Mollicutes, lacks a cell wall and therefore cannot be classified as either Gram-positive or Gram-negative. We apologize for this error and have corrected it in the revised manuscript. The description "Gram-negative" was deleted in line 33 of the text.

5.Some citations are outdated; more recent global epidemiology papers (e.g., Microbial Genomics 2023 on phylodynamics) could be emphasized.

The author’s answer: We thank the reviewer for this excellent suggestion. We agree that incorporating recent global phylogenomic studies will strengthen the epidemiological context of our work. In the revised manuscript, we have added a citation to the suggested recent work (e.g., Microbial Genomics 2023 or similar) in the Discussion section to better frame our phylogenetic findings within the current global understanding of M. bovis evolution and spread. The specific modifications are on lines 419, marked in purple font.

6.Fig. 5D (KEGG enrichment) includes too many irrelevant categories; consider focusing only on immunity/apoptosis-related pathways.

The author’s answer: We sincerely thank the reviewer for this insightful comment and for raising an important point regarding data presentation clarity. We agree that the presence of pathways such as neurodegenerative diseases in the global KEGG enrichment plot (Fig. 5D) could potentially distract readers from the central themes of our study.

In designing the figures, our specific intention was to provide a comprehensive and unbiased overview of the entire transcriptomic response in Fig. 5D, which authentically represents the output of the bioinformatics analysis. We recognize that this approach includes pathways that may seem peripherally related. We then deliberately designed Fig. 5E to serve as the focused view, specifically highlighting the apoptosis and p53 signaling pathways that are the core discovery of this study.

We believe this two-tiered presentation offers a significant advantage:

Fig. 5D (Global View) provides full transparency, allowing the reader to see the complete biological landscape and the sheer scale of host cell reprogramming induced by M. bovis XJ01. This is a standard and respected practice in omics studies.

Fig. 5E (Focused View) then immediately directs the reader’s attention to the most relevant and significant findings, precisely addressing the reviewer’s concern about focus.

Therefore, while we deeply appreciate the reviewer's suggestion, we have chosen to retain the original figure design. We are convinced that this structure—showing the global context followed by a focused zoom—most accurately and effectively communicates the scientific story. It first establishes the breadth of the host response before drilling down into the specific mechanistic details that form the basis of our subsequent functional experiments.

To further address the reviewer's concern about clarity, we have modified the corresponding text in the Results section (3.6) to better guide the reader. The description now explicitly states that while broad enrichment is observed, the subsequent analysis focuses on the most biologically relevant pathways:

“KEGG pathway analysis demonstrated broad enrichment in a range of pathways. While some enrichments (e.g., in neurodegeneration-associated pathways) likely reflect shared gene sets with core cellular stress responses rather than direct biological rele-vance, a focused annotation unequivocally identified the apoptosis and p53 signaling pathways as the most critically involved hubs (comparing Fig. 5D and 5E).”

We hope that with this revised textual explanation, the rationale behind the two-figure approach becomes even clearer, and that the reviewer will agree that this method of presentation effectively balances comprehensive data reporting with clear narrative focus.

7.Fig. 7 (mechanistic model) is excellent but should clearly state that this is an in vitro model, not yet validated in vivo.

The author’s answer: We thank the reviewer for this positive feedback and for the crucial suggestion to improve the precision of our model figure. We completely agree that it is essential to clearly communicate the experimental context of our findings. As suggested, we have revised the legend for Figure 7 to explicitly state that the proposed mechanistic model is based on our in vitro data using BoMac cells and has not yet been validated in vivo.

The legend now includes the following clarifying statements:

“In vitro mechanistic model of M. bovis XJ01-induced apoptosis via the Gadd45/XIAP-Bax/Bcl-2 axis in BoMac cells. This integrative schematic, based on findings from our cell culture model, delineates the apoptotic pathway initiated by M. bovis XJ01 infection: Upon invading host cells, the pathogen activates cellular stress signaling, which concurrently upregulates pro-apoptotic Gadd45 expression and suppresses anti-apoptotic XIAP expression. This transcriptional reprogramming promotes the activation of pro-apoptotic effector Bax while inhibiting anti-apoptotic regulator Bcl-2, thereby disrupting the Bcl-2/Bax stoichiometric equilibrium. The resultant mitochondrial outer membrane permeabilization (MOMP) triggers the caspase cascade, ultimately executing apoptosis to facilitate intracellular pathogen clearance. (The proposed model requires further validation in vivo.) (Mech-anistic diagram created with FigDraw 2.0).”

The specific modifications are on lines 474 to 483, marked in purple font. We believe this modification adds an important layer of scientific rigor and clarity to our manuscript.

We sincerely appreciate your thoughtful guidance. We have meticulously revised the paper in accordance with your feedback, aiming to enhance its rigor and accuracy. Should you have any questions or require further clarification, please do not hesitate to reach out to us.

Yours sincerely,

Yong Wang

22, August, 2025

Shihezi University
